# Transposon mutagenesis screen in *Klebsiella pneumoniae* identifies genetic determinants required for growth in human urine and serum

Jessica Gray[1,2†], Von Vergel L Torres[2†], Emily Goodall[2], Samantha A McKeand[1], Danielle Scales[1], Christy Collins[1], Laura Wetherall[1], Zheng Jie Lian[2], Jack A Bryant[1], Matthew T Milner[1], Karl A Dunne[1], Christopher Icke[2], Jessica L Rooke[2], Thamarai Schneiders[3], Peter A Lund[1], Adam F Cunningham[4], Jeff A Cole[1], Ian R Henderson[2]*

[1]Institute of Microbiology and Infection, University of Birmingham, Birmingham, United Kingdom; [2]Institute for Molecular Bioscience, University of Queensland, Brisbane, Australia; [3]Division of Infection Medicine, University of Edinburgh, Edinburgh, United Kingdom; [4]Institute of Immunology and Immunotherapy, University of Birmingham, Birmingham, United Kingdom

*For correspondence: i.henderson@imb.uq.edu.au

†These authors contributed equally to this work

Competing interest: The authors declare that no competing interests exist.

**Abstract** *Klebsiella pneumoniae* is a global public health concern due to the rising myriad of hypervirulent and multidrug-resistant clones both alarmingly associated with high mortality. The molecular mechanisms underpinning these recalcitrant *K. pneumoniae* infection, and how virulence is coupled with the emergence of lineages resistant to nearly all present-day clinically important antimicrobials, are unclear. In this study, we performed a genome-wide screen in *K. pneumoniae* ECL8, a member of the endemic K2-ST375 pathotype most often reported in Asia, to define genes essential for growth in a nutrient-rich laboratory medium (Luria-Bertani [LB] medium), human urine, and serum. Through transposon directed insertion-site sequencing (TraDIS), a total of 427 genes were identified as essential for growth on LB agar, whereas transposon insertions in 11 and 144 genes decreased fitness for growth in either urine or serum, respectively. These studies not only provide further knowledge on the genetics of this pathogen but also provide a strong impetus for discovering new antimicrobial targets to improve current therapeutic options for *K. pneumoniae* infections.

## eLife assessment

This **valuable** study is of relevance for those interested in the mechanism required for infections of humans by *Klebsiella pneumoniae*. The authors apply TraDIS (high-density TnSeq) to *K. pneumoniae* with the goal of identifying genes required for survival under various infection-relevant conditions and the gene sets identified, together with the raw sequence data, will be resources for the Klebsiella research community. The evidence to support the lists of essential and conditionally-essential genes is **convincing**. The study provides strong evidence that some genes are conditionally essential in urine because of iron limitation, but there is less mechanistic insight for genes that are conditionally essential in serum.

## Introduction

*Klebsiella pneumoniae* are Gram-negative bacteria found ubiquitously in natural and man-made environments (*Bagley, 1985*; *Holt et al., 2015*; *Podschun and Ullmann, 1998*). In immunocompromised patients, and those with comorbidities such as diabetes or alcoholism, *K. pneumoniae* can colonize multiple body sites causing a diverse array of infections ranging from life-threatening pneumonia, urinary tract infections, and bacteremia (*Ko et al., 2002*; *Gonçalves Barbosa et al., 2022*; *Tumbarello et al., 2015*; *Clegg and Murphy, 2016*; *Paczosa and Mecsas, 2016*). *K. pneumoniae* is a frequently reported multidrug-resistant pathogen isolated from nosocomial settings. Such infections have generally higher mortality rates and require longer treatment regimens and hospital stays, placing a significant burden on healthcare providers (*GBD 2019 Antimicrobial Resistance Collaborators, 2022*; *Pham et al., 2023*). Of most concern is resistance to β-lactam antibiotics such as penicillin, aztreonam, and extended-spectrum cephalosporins, by horizontally acquired plasmid-mediated β-lactamases (*Navon-Venezia et al., 2017*). While combined antibiotic therapies have been deployed in the clinic, the emergence of pandrug-resistant strains has created an urgency to develop new and efficient approaches to treat *K. pneumoniae* infections (*Garbati and Al Godhair, 2013*; *Campos et al., 2016*; *Rojas et al., 2017*). The identification of genetic determinants essential for growth and those required for survival of *K. pneumoniae* in vivo may help elucidate novel therapeutic targets (*Cain et al., 2020*; *Barquist et al., 2016*).

Recent large-scale phylogenetic investigations have suggested that distinct lineages of *K. pneumoniae* exist in clinical and environmental settings (*Wyres et al., 2020*). Furthermore, two clinical pathotypes of *K. pneumoniae* have emerged: classical and hypervirulent. While the full repertoire of genetic factors required for *K. pneumoniae* virulence remains unclear, various studies have highlighted the importance of genes involved in adhesion, iron acquisition, capsulation, and cell envelope biogenesis (*Marques et al., 2019*). Nevertheless, for both classical and hypervirulent *K. pneumoniae* strains, the ability to colonize the urinary tract and survive in the bloodstream are essential pathogenic traits and are dependent on a large and diverse range of virulence factors (*Bengoechea and Sa Pessoa, 2019*). Indeed, previous studies demonstrated that lipopolysaccharide (LPS) (O-) (9 serotypes) and capsular polysaccharide (*cps*) (K-) antigens (79 serotypes) contribute to the resistance of serum-mediated killing and antiphagocytosis (*Choi et al., 2020*; *Martin and Bachman, 2018*). However, prior genetic screens have revealed that serum resistance in Gram-negative bacteria is an intricate phenotype defined by a multitude of factors (the serum resistome) (*Short et al., 2020*; *Phan et al., 2013*; *McCarthy and Taylor, 2022*; *Ondari et al., 2019*; *Subashchandrabose et al., 2016*). A recent study of *K. pneumoniae* identified 93 different genes associated with serum resistance across four distinct strains but found that only three genes (*lpp*, *arnD*, and *rfaH*) played a role in all strains. These data hint that multiple mechanisms of serum resistance may exist in *Klebsiella* lineages. In contrast, the full repertoire of genes required for growth of *K. pneumoniae* in the urinary tract (the urinome) has not been defined.

To define the genetic basis of the *K. pneumoniae* urinome and serum resistome, we used transposon insertion sequencing (TIS), also referred to as transposon directed insertion-site sequencing (TraDIS). TraDIS is a genome-wide screening technique that has been widely used to identify genes essential for bacterial growth and to discover conditionally essential genes under physiologically relevant conditions (*Goodall et al., 2018*; *Fabian et al., 2021*; *Wong et al., 2016*; *Ramage et al., 2017*; *DeJesus et al., 2017*; *Ebert et al., 2013*; *Hardy et al., 2021*; *Armbruster et al., 2017*; *Roth et al., 2022*; *Winkle et al., 2021*; *Goodall et al., 2021*). In this study, we generated a highly saturated transposon library within *K. pneumoniae* ECL8, a member of the K2-ST375 phylogenetic lineage that includes hypervirulent clones of epidemiological significance known to cause infections in relatively healthy subjects predominantly in Asia (*Hoashi et al., 2019*). We identified the repertoire of genes essential for growth in laboratory conditions, in pooled human urine, and in human serum. Selected fitness genes were validated for growth in pooled urine or serum by generating single-gene deletion mutants. Our study provides insight into the molecular mechanisms that enable *K. pneumoniae* to survive in vivo and cause disease.

## Results and discussion

### Generation and sequencing of a *K. pneumoniae* strain ECL8 transposon input library

A *K. pneumoniae* ECL8 mini-Tn5 mutant library composed of >1 million mutants was constructed and subsequently subjected to TraDIS. The resulting reads were mapped to the reference genome that consists of an ~5.3 Mb chromosome and a single 205 kb plasmid (EMBL Accessions: HF536482.1 and HF536483) (*Figure 1A and B*). The gene insertion index scores (IIS) of the two technical replicates of the *K. pneumoniae* ECL8 TraDIS library (KTL1 and KTL2) were highly correlated with each other ($R^2$=0.955), demonstrating a high level of reproducibility (*Figure 1—figure supplement 1A*). Therefore, sequence reads for both replicates were combined for downstream analysis and essential gene prediction. Through iterative rounds of sequencing, we determined that the library was sequenced to near saturation as the discovery of unique reads plateaued at ~2 million reads (*Figure 1—figure supplement 1B*). This library contained 554,834 unique genome-wide transposon insertion sites and 499,919 of these insertions map within annotated coding sequence (CDS) boundaries. This high level of transposon coverage equates to an average transposon insertion every 9.93 bp throughout the genome, equivalent to an insertion approximately every four codons. A de Brujin graph visualized in the genome assembly package Bandage revealed that the sequencing coverage of the plasmid was only 1.33-fold higher than that of the chromosome indicating that the plasmid is likely present as a single copy (*Figure 1—figure supplement 2*; *Wick et al., 2015*). *Table 1* summarizes the data relating to the ECL8 libraries.

### Essential chromosomal genes in *K. pneumoniae* ECL8

To classify a gene as essential or non-essential, the number of transposon insertions within each gene on the chromosome and plasmid was normalized for gene length and this value was denoted the gene IIS (*Figure 1C*). IIS for the *K. pneumoniae* genome followed a bimodal distribution, as has been previously described (*Figure 1—figure supplement 3*; *Goodall et al., 2018*; *Langridge et al., 2009*). The probability of a gene being essential was calculated by determining the likelihood of each given IIS as belonging to the essential or non-essential mode. The ratio of the IIS values was denoted as the log-likelihood ratio ($\log_2$-LR), a metric previously used to categorize genes into essential or non-essential groupings (*Goodall et al., 2018*; *Langridge et al., 2009*). To limit the number of false-positive hits identified, genes were classified as essential only if they were 12 times more likely to be within the essential, rather than the non-essential mode. Based on the above criteria, 373 chromosomal genes were assigned a $\log_2$-LR of less than –3.6 and were therefore classified as essential for growth on solid Luria-Bertani (LB) medium supplemented with kanamycin (*Figure 1D*). It should be noted that non-CDSs such as tRNA and rRNA were excluded from this analysis. Most genes (4551) were classified as non-essential (*Figure 1E*). A subset of 241 genes were located between these two modes with their essentiality mode being deemed 'unclear' (*Figure 1F*). A complete list for all *K. pneumoniae* ECL8 bimodal essentiality categorization and their associated statistical significance metrics are listed in *Figure 1—source data 1*.

To understand the essential gene functions, we evaluated their COG (cluster of orthologous) categories; a metric that predicts functional classification against a database of known proteins. We applied a $\log_2$ COG enrichment index to identify COG categories that were enriched among the essential genes in contrast to the wider genome eggNOG (v5.0) orthology data (*Figure 1—figure supplement 4*; *Huerta-Cepas et al., 2019*). We identified no essential genes within the COG categories for chromatin structure and dynamics (B), cell motility (N), and secondary structure (Q) suggesting genes within these categories are not required or redundant for *K. pneumoniae* growth in the utilized nutrient-rich liquid medium. However, genes involved in signal transduction (T) and inorganic ion transport/metabolism (P) were depleted, while genes involved in translation (J), cell cycle control (D), and co-enzyme metabolism (H) were the most highly enriched. This correlates with enriched COG categories shared among essential genes in other members of the Proteobacteria such as *Salmonella enterica* Typhimurium, *S. enterica* Typhi, and *Caulobacter crescentus* and serves as an internal control for the validity of the library (*Christen et al., 2011*; *Grazziotin et al., 2015*; *Barquist et al., 2013*).

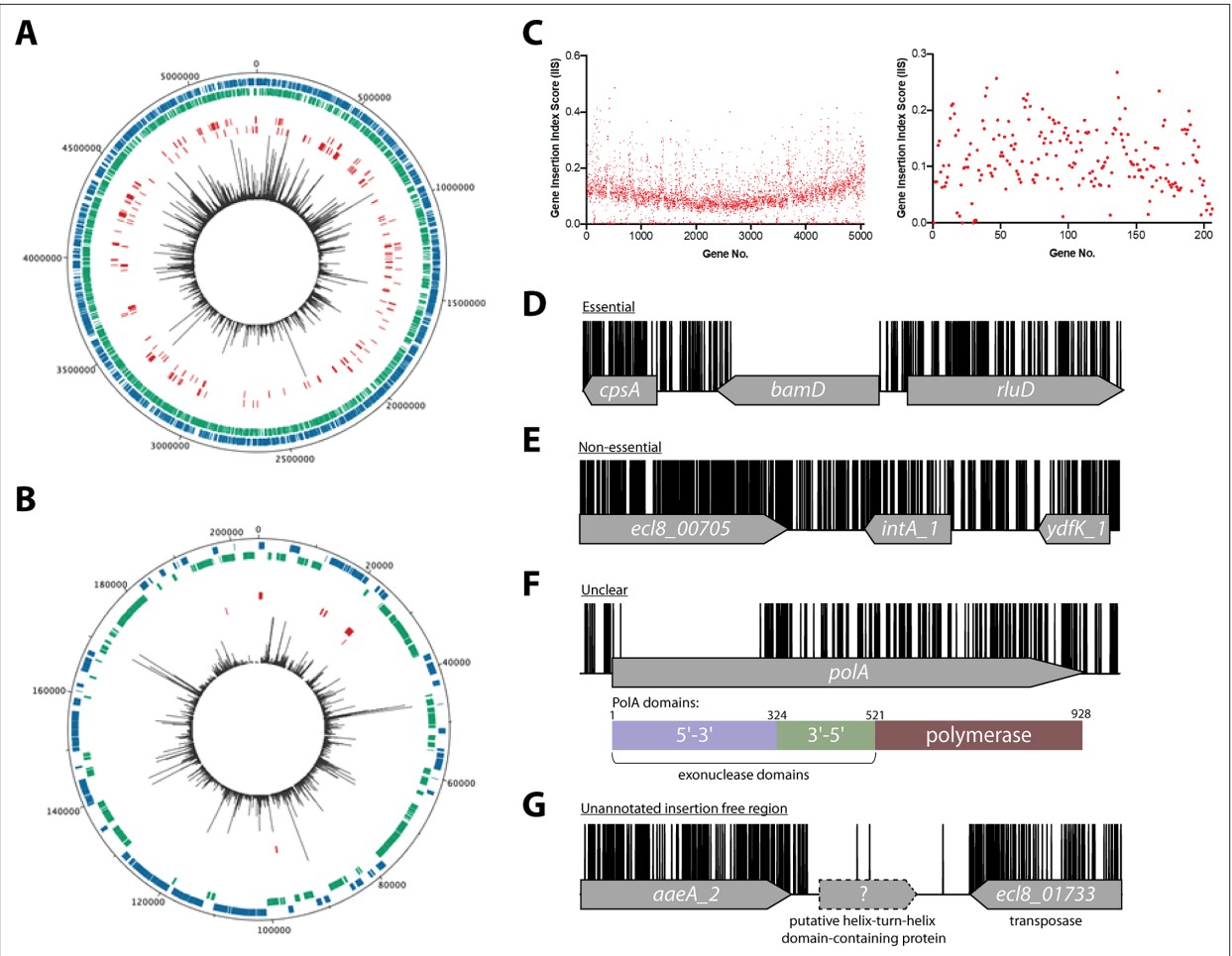

**Figure 1.** Overview of ECL8 transposon directed insertion-site sequencing (TraDIS) data: mapped insertions, insertion index profile, and example gene insertion plots. Insertions are illustrated on the (**A**) chromosome and (**B**) plasmid, respectively. The outermost track displays the length of the ECL8 genome in base pairs. The subsequent two inner tracks correspond to coding sequences (CDSs) on the sense (blue) and antisense (green) DNA strands, respectively. Putative essential CDSs are highlighted in red. The innermost track (black) corresponds to the location and read frequency of transposon sequences mapped successfully to the *K. pneumoniae* ECL8 genome. Plot generated using DNAPlotter. (**C**) Gene insertion index scores (IIS) of the *K. pneumoniae* ECL8 TraDIS library mapped in order of genomic annotation of the *K. pneumoniae* ECL8 (left) chromosome and (right) plasmid. Example transposon insertion profiles categorized into essential, non-essential, and unclear: (**D**) an essential gene – *bamD*, an essential outer membrane factor for β-barrel protein assembly; (**E**) a non-essential gene – *int*_A1, a redundant (several copies) integrase required for bacteriophage integration into the host genome; (**F**) an 'unclear' gene – *polA,* an essential gene in prokaryotes required for DNA replication but showed requirement for the N-terminal 5'–3' exonuclese domain; and (**G**) an insertion-free region suggestive of an unannotated open reading frame (ORF). Transposon insertion sites are illustrated in black and capped at a maximum read depth of 1.

The online version of this article includes the following source data and figure supplement(s) for figure 1:

**Source data 1.** Essential gene table ECL8.

**Source data 2.** Insertion-free regions (IFRs) within ECL8 (Luria-Bertani [LB]).

**Figure supplement 1.** Analysis of TraDIS data.

**Figure supplement 2.** Sequencing depth of the *K. pneumoniae* ECL8 plasmid (black) relative to the *K. pneumoniae* ECL8 chromosome (green).

**Figure supplement 3.** The frequency distribution of insertion index scores.

**Figure supplement 4.** The COG (cluster of orthologous) enrichment index comprising 373 genes classified as essential in *K. pneumoniae* ECL8.

**Figure supplement 5.** Mathematical simulation (10⁵ instances) of random transposon insertion events under the null model of random insertion previously described (***Podschun and Ullmann, 1998***).

**Figure supplement 6.** Sequencing depth required to sample a given proportion of the *K. pneumoniae* transposon directed insertion-site sequencing (TraDIS) library.

**Table 1.** Summary of transposon-containing sequence reads and unique insertion points (UIPs) mapped to the *K. pneumoniae* ECL8.

| Sample | Mapped reads after QC | Genome-wide UIPs | CDS UIPs | Pearson correlation of replicates |
|---|---|---|---|---|
| ECL8 Input libraries (LB) | | | | |
| KTL1 | 5,573,710 | 409,069 | 367,230 | *Figure 1—figure supplement 1A* |
| KTL2 | 2,854,389 | 400,915 | 361,272 | |
| KTL (combined) | 8,429,782 | 554,834 | 499,919 | |
| Urine output libraries | | | | |
| LB | 4,324,125 | 4,324,125 | 403,381 | *Figure 4—figure supplement 1* |
| Urine | 4,839,277 | 4,839,277 | 389,251 | |
| Serum output libraries | | | | |
| Serum | 4,729,032 | 115,328 | 101,348 | *Figure 6—figure supplement 2* |
| Heat-inactivated serum | 3,565,433 | 284,761 | 254,735 | |
| 180 min serum exposed | 1,481,732 | 133,790 | 121,316 | *Figure 7—figure supplement 1* |

## 'Essential' plasmid genes

The identification of 11 genes on the *K. pneumoniae* ECL8 plasmid that met the criteria to be defined as essential was an unexpected observation (*Table 2*). As previously noted, the bacterial copy number of the plasmid was approximated to be one. The average IIS of a gene located on the plasmid and the chromosome was comparable, 0.16 and 0.11, respectively. Due to the presence of a single plasmid per bacterial cell, and an observed even distribution of IISs for genes located on the plasmid, the lack of insertions in the 11 genes is unlikely to be due to gene dosage effects arising from the presence of multiple copies of the plasmid. Notably, the 11 plasmid-borne 'essential' genes included *repB*, which is required for plasmid replication. Therefore, these genes are likely to be required for plasmid

**Table 2.** *K. pneumoniae* ECL8 plasmid-borne genes computationally deemed essential.

| Locus tag | Gene | Function* |
|---|---|---|
| ecl8_05075 | *repB*_1 | RCR (rolling circle replication) plasmid protein |
| ecl8_05092 | – | SidC homolog |
| ecl8_05094 | – | DUF2509 family protein |
| ecl8_05096 | – | NB-ARC domain-containing protein |
| ecl8_05104 | *rcsA*_2 | DNA-binding transcriptional activator RcsA |
| ecl8_05105 | *yedA*_2 | Putative transporter |
| ecl8_05106 | – | DUF2695 domain-containing protein |
| ecl8_05170 | – | DUF305 domain-containing protein |
| ecl8_05205 | – | NADPH-dependent preQ0 reductase |
| ecl8_05272 | – | Transposase |
| ecl8_05279 | *sopB* | Control of plasmid partitioning |

*Function derived from the top hit using NCBI blastN (https://blast.ncbi.nlm.nih.gov/Blast.cgi).

replication and stability and are unlikely to be essential for growth. This hypothesis is supported by previous observations that *K. pneumoniae* remains viable when large indigenous virulence plasmids are cured from the bacterium (*Tang et al., 2010*; *Rodríguez-Medina et al., 2020*; *Yang et al., 2021*).

## Insertion-free regions

Using a highly saturated transposon library it is possible to not only define essential genes, but also define chromosomal regions lacking transposons that are large enough not to have occurred by chance. Previously, we described a geometric model to predict statistically significant regions of the genome that were free from transposon insertions (*Goodall et al., 2018*). This model indicated that with the insertion frequency calculated for this library, for a genome of 5.3 Mb, an insertion-free region (IFR) of 145 bp or greater was statistically significant (*Figure 1—figure supplement 5*). The size of IFRs that are statistically significant for two previously published TraDIS libraries are plotted for comparison (*Goodall et al., 2018*; *Langridge et al., 2009*). A total of 667 genomic regions with IFRs≥145 bp were identified, many which correspond to the 380 essential genes identified through the bimodal analysis described earlier.

The remaining IFRs can be explained in several ways. First, as noted previously (*Goodall et al., 2018*), genes that encode proteins with essential domains are often excluded from the bimodal analysis described above as much of the gene contains transposon insertions. For example, the *polA* gene was classified as 'unclear' but contained, barring a single transposon insertion event, an 826 bp IFR. This IFR corresponds to 275 amino acids that are predicted to encode the 5'–3' exonuclease function of the protein (*Figure 1F*). This region has also been shown to be required for the growth of *Escherichia coli* in laboratory conditions (*Joyce and Grindley, 1984*; *Gallagher et al., 2020*). In contrast, but as noted for *E. coli polA*, multiple transposon insertions are present in the remaining portion of the gene, which confers the 3'–5' proofreading and polymerase activity. Aside from those genes already identified by the bimodal analyses described above, our analyses revealed 54 additional genes with statistically significant IFRs, which indicates that they are likely essential for growth under the conditions tested here. Thus, our data suggest *K. pneumoniae* contains at least 427 essential genes, which we define as the ECL8 curated essential gene list.

An explanation for the remaining IFRs can be derived from traditional bioinformatic approaches to genome annotation. Originally, genome annotation protocols overlooked small open reading frames

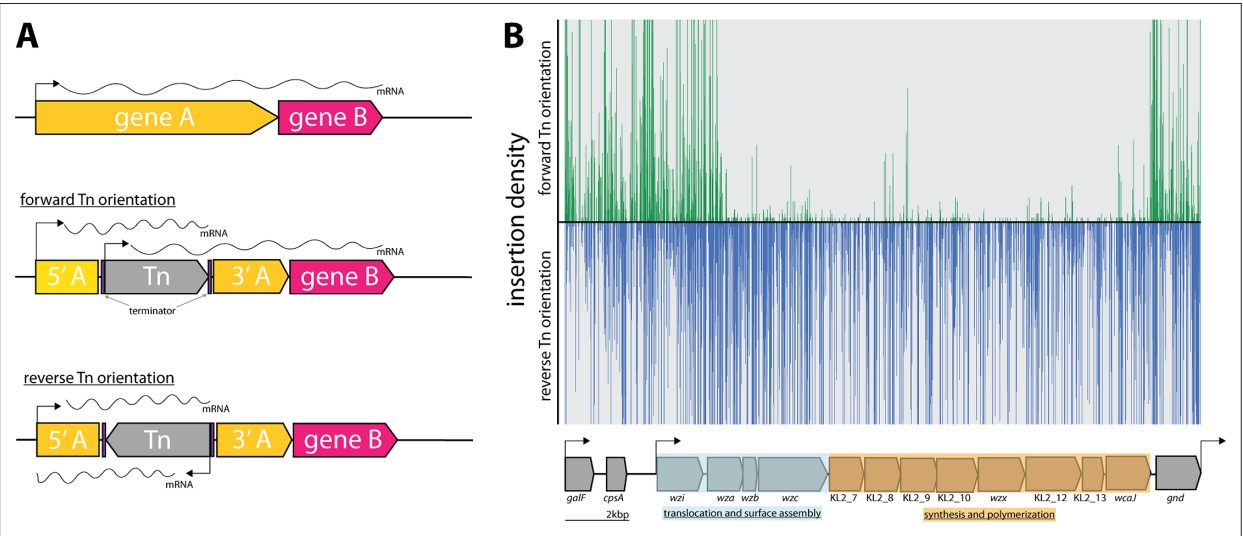

**Figure 2.** Directional insertion bias of transposon (Tn) into the capsular polysaccharide (*cps*) operon. (**A**) Schematic representing the Tn orientation and effect on downstream transcription. The utilized Tn transposon is flanked by terminator sequences (purple) and is shown inserted in gene A (yellow) of a hypothetical two-gene transcription unit AB in the forward or reverse orientation. In the forward orientation transcription of gene B (pink) is expected to occur from the promoters of 5' of gene A or the internal Tn5 but polycistronic mRNA differs in length due to the attenuation by the terminators. (**B**) Transposon insertions mapping to the K2 capsular operon of *K. pneumoniae* ECL8. Transposon insertions are configured in the forward orientation (green), and reverse orientation (blue) and insertion densities are capped at a maximum read depth of 50. The operon structure of the K2 capsular genes consisting of three promoters driving the expression of three unidirectional polycistronic transcripts is depicted.

(ORFs) because (i) they only consider genes that code for proteins >100 amino acids, and (ii) define genes based on sequence homology to those that have been previously annotated (*Warren et al., 2010*; *Mat Sharani and Firdaus-Raih, 2019*; *Miravet-Verde et al., 2019*). Interestingly, the methodology used here revealed 59 regions ≥145 bp that lacked transposon insertion sites and did not map to within annotated CDS boundaries (*Figure 1G*). The applied approach is conservative and does not include IFRs that overlap with the 5' or 3' region of any annotated CDS. Forty-six of these regions contain potential ORFs that could encode proteins through blastX identification. These putative unannotated ORFs were not characterized further within this study but might represent essential genes. This study emphasizes the untapped potential of TraDIS datasets to identify previously uncharacterized, essential, small CDSs, and promoter elements that are not immediately obvious by traditional and computational genomic annotation methods. A complete list of IFRs, within annotated ORFs or in intergenic regions, and an ECL8 curated essential gene list is found in *Figure 1—source data 2*.

## Directional insertion bias of Tn5 transposon into *cps* operon

As reported previously (*Goodall et al., 2018*; *Hutchison et al., 2019*), due to the design of the transposon used in this study, transcription of polycistronic operons can be affected by the insertion orientation of the transposon (*Figure 2A*). Directional insertion bias of the transposon was noted within a large genomic region of ~14 kb encoding the virulence-associated *cps* operon (*Figure 2B*). The bioinformatic webtool 'Kaptive' determined with high confidence the *K. pneumoniae* ECL8 *cps* operon encodes for a KL2/K2 capsule with 18/18 of the expected genes (*Figure 7—source data 1*; *Lam et al., 2022*).

Notably, transposon insertions were poorly tolerated in the forward Tn orientation of the DNA strand. While bias for Tn insertions has previously been described in individual genes (e.g. ribosomal RNA genes), this study is the first to describe such a large bias across an operon (*Goodall et al., 2018*; *Christen et al., 2011*). However, the basis for this bias is not fully understood. One potential explanation for such bias might be the presence of a gene encoding a toxic small RNA (tsRNA). Typically, tsRNAs are usually present in the 5' or 3' untranslated regions of genes and act as regulators of gene expression at a post-transcriptional level, but due to their small size, typically <500 nucleotides, they are very difficult to predict (*Pain et al., 2015*; *Waters and Storz, 2009*). However, this hypothesis does not explain why transposon insertions are uniformly absent for the opposite orientation of the transposon and not isolated to 'hotspots' in the 5' and 3' untranslated regions. An alternative hypothesis is that the reverse orientation expression of the capsular operon might result in the accumulation of intracellular capsular intermediates that may be toxic if left to accumulate in the periplasm. This finding may hint to a more cryptic regulation mechanism of the *cps* operon with implications for other similar *Klebsiella* capsular serotypes and species.

## Comparison of essential genes to other *K. pneumoniae* transposon libraries

Transposon libraries have previously been generated in other *K. pneumoniae* strains, however these studies primarily focused on identifying genes that are required for a specific phenotype, i.e., antibiotic stress or capsular mutants, and therefore they lack a definitive list of essential genes for in vitro growth (*Jana et al., 2017*; *Dorman et al., 2018*; *Jung et al., 2019*; *Paczosa et al., 2020*; *Ma et al., 2021*). To benchmark our library, we compared our data to previously published studies that did report *K. pneumoniae* essential gene lists: KPNIH1 (ST258), RH201207 (ST258), and ATCC 43816 (ST493) (*Ramage et al., 2017*; *Bruchmann et al., 2021*). As shown in *Figure 3*, the ECL8 strain is an acceptable representative for the *K. pneumoniae* KpI phylogroup as demonstrated by the phylogenetic and ANI (average nucleotide identity) analyses in the context of *K. pneumoniae* strains previously used or 'common' nosocomial lineages (*Gorrie et al., 2022*; *Feldman et al., 2019*; *Lin et al., 2008*; *Chhibber et al., 2008*). Furthermore, our transposon mutant library is among the most saturated *K. pneumoniae* libraries reported to date, permitting accurate determination of gene essentiality and delineating between real and stochastic effects. The number of essential genes among the four strains ranged from 434 to 642; a complete list and comparison of our curated gene list against the reported gene lists of KPNIH1, RH201207, and ATCC 43816 is found in *Figure 3—source data 1*. We speculate that the higher number of essential genes predicted in the KPNIH1 gene list is due to a lower transposon density, as this is associated with an increased likelihood of false-positive results

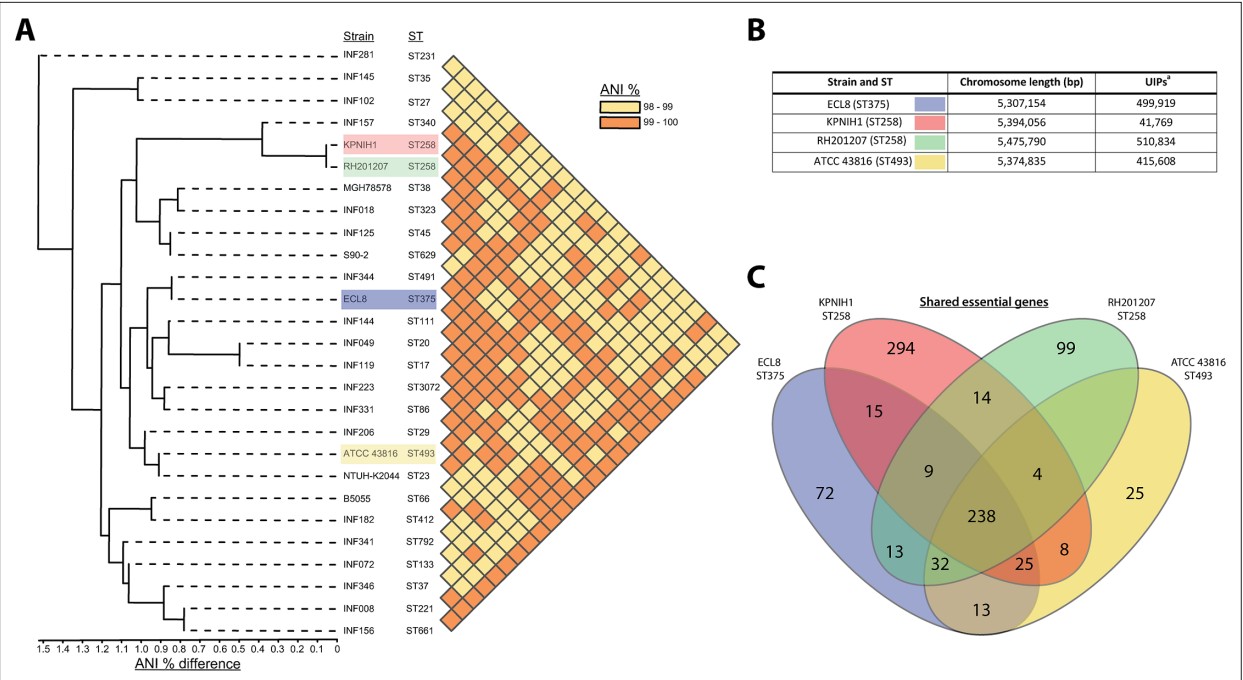

**Figure 3.** Phylogenetic context of *K. pneumoniae* ECL8 and comparison to previously reported essential gene lists. (**A**) Phylogenetic tree and average nucleotide identity (ANI) analysis generated using the Integrated Prokaryotes Genome and Pan-genome Analysis (IPGA) webserver demonstrating the phylogenetic context of *K. pneumoniae* ECL8 compared to other previously published *K. pneumoniae* strains or isolates belonging to '21 "common" lineages' of nosocomial origin by other groups. The ANI is a similarity index metric between a given pair of genomes applicable to prokaryotic organisms. A cutoff score of >95% typically indicates they belong to the same species. (**B**) Tabular comparison of *K. pneumoniae* strains listing genome size and number of unique insertion points mapped (**C**) Venn diagram depicting the shared and unique genes required for growth in nutrient-rich media (i.e. Luria-Bertani [LB]). Complete list of genomes, gene comparisons, and exclusions lists can be found in *Figure 3—source data 1*.

The online version of this article includes the following source data for figure 3:

**Source data 1.** Essential gene comparison against ECL8.

(*Chao et al., 2016*). This might be due in part to differences in the techniques used to construct the libraries; here, we used a mini-Tn5 transposon which showed no bias in insertion whereas others have used the TnSeq method relying on the Himar I Mariner transposon, which has a requirement for a TA dinucleotide motif (*Lampe et al., 1996*). Differences in gene essentiality might also be due to inherent genomic differences or due to differences in experimental methodology, computational approaches, or the stringency of analysis used to categorize these genes. While further validation experiments will be required to determine whether specific genes are essential, 57% of the essential genes identified in *K. pneumoniae* ECL8 were essential in the other three strains. Thus, together these data indicate that our transposon library is sufficiently dense and representative of *K. pneumoniae* (KpI phylogroup) to be used for further studies.

## Identification of genes required for growth in human urine

TIS has become a formidable tool for rapid genome-scale screens to link genotype to phenotype. As the urinary tract is a major site for *K. pneumoniae* infection, we used our transposon mutant library to identify genes required for growth in urine, thus enabling us to define the urinome of *K. pneumoniae*. Cultures were grown until late stationary phase (12 hr) followed by two subsequent 12 hr passages before sequencing (*Figure 4A*). This approach eliminated mutants that were unable to grow in urine while allowing those capable of growth to flourish. DNA from test and control samples was harvested, sequenced, and analyzed as before (*Figure 1—figure supplement 6*). As the gene IISs for the library passaged in both LB and urine were highly correlated with respective biological replicates of 0.8483 and 0.9078 $R^2$, replicates were combined for downstream analyses (*Figure 4—figure supplement 1*). The data for all *K. pneumoniae* genes and their associated statistical significance metrics are listed in *Figure 4—source data 1*.

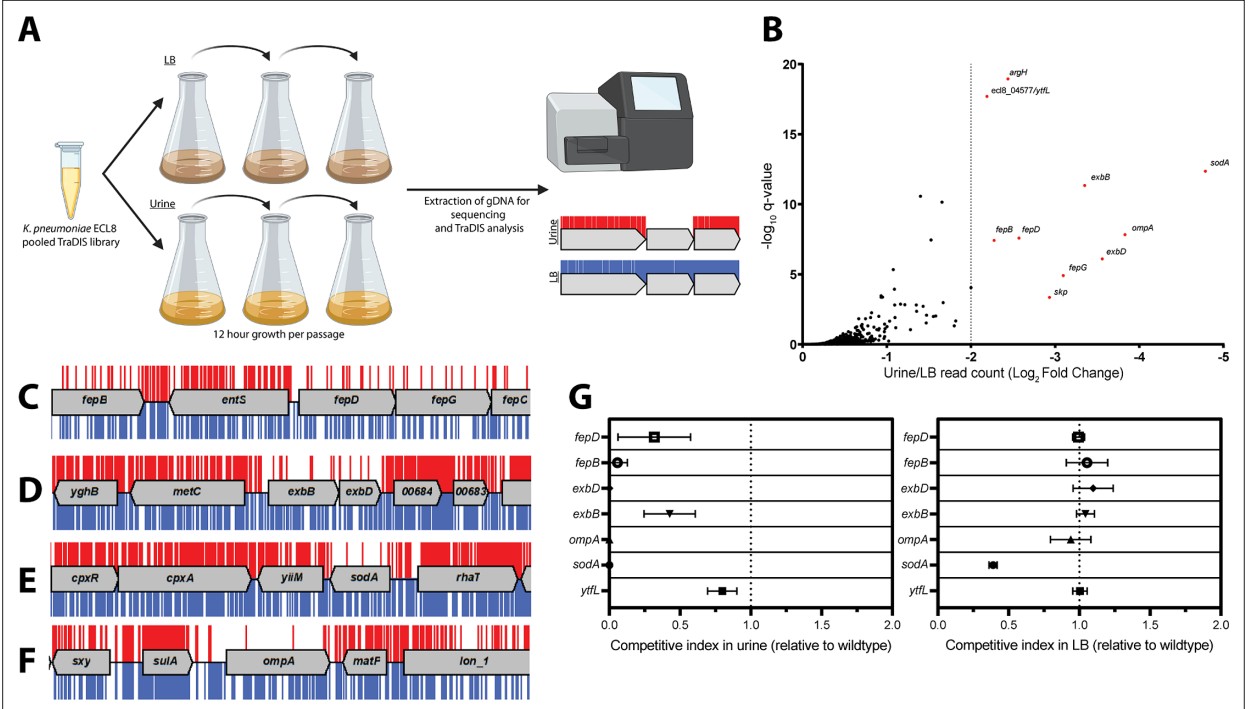

**Figure 4.** Overview and validation of ECL8 fitness factors for growth in pooled human urine. (**A**) Schematic of the experimental design used to identify genes that provide a fitness advantage for *K. pneumoniae* ECL8 growth in pooled human urine. The *K. pneumoniae* ECL8 library was inoculated into either 50 mL of Luria-Bertani (LB) or 50 mL of urine and incubated at 37°C with 180 RPM shaking for 12 hr. The library was passaged into 50 mL of fresh LB or pooled human urine at an initial $OD_{600}$ of 0.05 two subsequent times. A 1 mL sample normalized to an $OD_{600}$ of 1 from each culture was processed for genome extraction and multiplexed sequencing using an Illumina MiSeq. (**B**) Log₂ fold change (log₂FC) of the read count for each *K. pneumoniae* ECL8 gene when passaged in urine relative to an LB control. Genes highlighted in red satisfy a stringent applied threshold (log₂FC>–2, Q-value≤0.05). The Q-value is the p-value that has been adjusted for the false discovery rate for each gene. For brevity, only genes with a log₂FC≥0 are illustrated. Selected transposon insertion profiles of genes identified as advantageous for growth in urine: (**C**) *fepB*, *fepD*, *fepG*, (**D**) *exbB*, *exbD*, (**E**) *sodA*, and (**F**) *ompA*. These genes exhibited a significant loss of transposon insertions following growth in urine (red) in comparison to LB broth (blue). A 5 kb genomic region including the gene is illustrated. Reads are capped at a maximum depth of 1. (**G**) The fitness of gene replacement mutants relative to wild-type (WT) *K. pneumoniae* ECL8 in either LB medium or urine. The relative competitive index of single-gene replacement mutants after 12 hr passages 3× in either LB medium or urine. A relative fitness of one would indicate comparable fitness to WT. The mean (*n*=3) is plotted (±1 SD).

The online version of this article includes the following source data and figure supplement(s) for figure 4:

**Source data 1.** Essential gene table ECL8 (urine).

**Figure supplement 1.** The Pearson correlation coefficient (R²) of gene insertion index scores (IIS) for two sequenced biological replicates of the *K. pneumoniae* ECL8 transposon directed insertion-site sequencing (TraDIS) library following three 12 hr passages in (blue) Luria-Bertani (LB) broth, or (red) pooled human urine.

Ten genes that satisfied a stringent threshold of a log₂ fold change (log₂FC)>–2 and a Q-value≤0.05, relative to the LB control, were considered as advantageous for *K. pneumoniae* ECL8 growth in urine (**Figure 4B**). The transposon insertion profiles of these genes showed a marked decrease in overall insertions following passaging in urine when compared to the LB control (**Figure 4C–F**), suggesting loss of these genes confers a fitness defect. These included genes encoding the superoxide dismutase SodA, the structural outer membrane protein OmpA (**Pain et al., 2015**), the periplasmic chaperone Skp (**Mas et al., 2019**), and five proteins (*fepB*, *fepD*, *fepG*, *exbB*, and *exbD*) associated with iron acquisition (**Caza and Kronstad, 2013**). The remaining two genes (*ytfL* and *argH*) play key roles in controlling cytoplasmic cadaverine and putrescine concentrations and amino acid biosynthesis, respectively (**Iwadate et al., 2021**; **Xu et al., 2000**). The proper regulation of cadaverine and putrescine maintains intracellular pH concentrations and reduces oxidative damage to proteins and DNA and may be in response to acidic conditions in urine known to be unconducive for colonization of uropathogens (**Chambers and Lever, 1996**; **Chaturvedi et al., 2012**).

To experimentally validate our observations, seven independent mutants were constructed by allelic replacement of the chromosomal gene with a kanamycin (*aph*) cassette. To determine whether

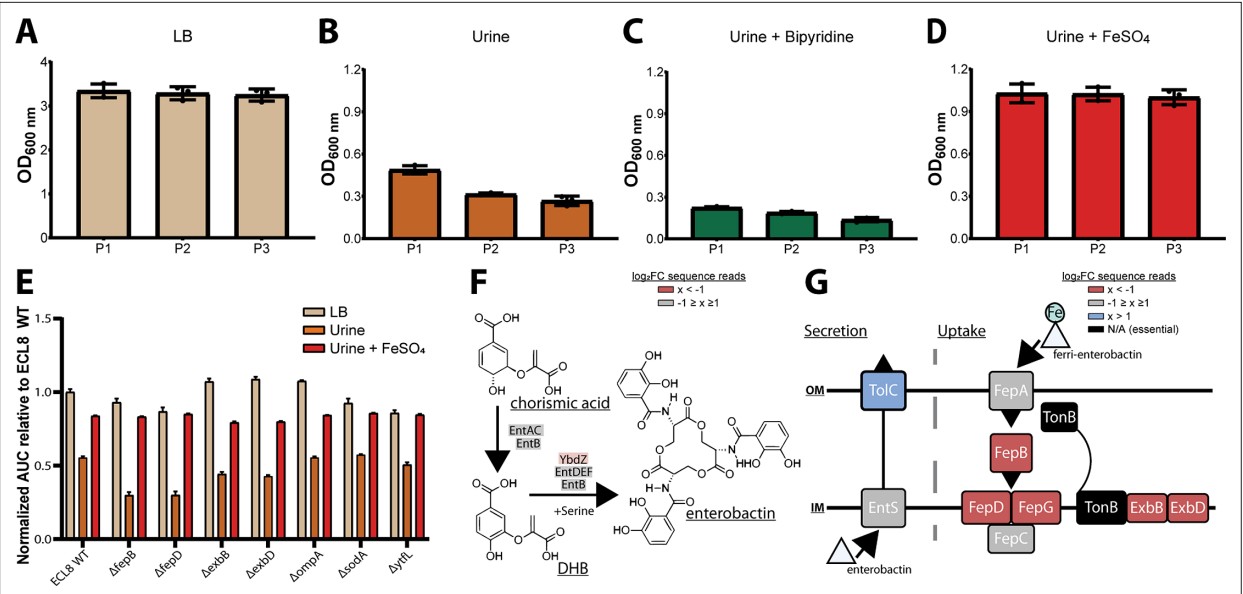

**Figure 5.** Growth of the *K. pneumoniae* transposon directed insertion-site sequencing (TraDIS) library following passaging in Luria-Bertani (LB) and urine and schematic diagrams of enterobactin synthesis, secretion, and uptake. The $OD_{600}$ of the *K. pneumoniae* TraDIS library following 12 hr of growth (P1) and two sequential 12 hr passages (P2 and P3). The library was passaged into fresh medium (**A**) LB or (**B**) urine to an initial $OD_{600}$ of 0.05. To determine the effect of iron supplementation and depletion, urine was supplemented with exogenous iron (**C**) 100 µM $FeSO_4$ or an iron chelator (**D**) 100 µM 2,2-dipyridyl. The average $OD_{600}$ of three biological replicates for each time point is plotted (±) 1 SD. (**E**) Area under curve comparative analysis ($OD_{600}$ vs. time) of *K. pneumoniae ECL8* and mutants grown in LB, urine, or urine supplemented with 100 µM $FeSO_4$ grown for 16 hr with 180 RPM shaking. The average of three biological replicates is plotted for each condition with error bars representing SD. (**F**) Simplified schematic of the enterobactin synthesis pathway. YbdZ, a co-factor of EntF for the terminal steps for enterobactin synthesis, depicted in light red had a $log_2$ fold change ($log_2FC$) sequence read value of −1.58, suggesting this gene conferred an overall fitness advantage for growth in urine. (**G**) Schematic representation of enterobactin secretion and uptake. The TonB transport system is present in Gram-negative bacteria and is required to transport Fe-bound enterobactin through the outer (OM) and inner membrane (IM) to the cytosol where it can be utilized. Based on $log_2FC$ sequence read value, loss of TolC (blue) was beneficial for growth, relative to an LB control. Loss of proteins, colored in red, had $log_2FC$ sequence read values <−2 suggesting they confer a fitness advantage for growth in urine. Proteins depicted in gray were genes that had $log_2FC$ values that ranged from −1 to 1 exposed to urine relative to an LB control. Genes depicted in black were essential and had no determinable $log_2FC$ value.

*K. pneumoniae* strains lacking the genes identified in this screen were truly less fit in urine compared to the parent strain, the wild-type (WT) ECL8 and individual isogenic mutant strains were inoculated into urine or LB medium at a 1:1 ratio. These cultures were sequentially passaged, and the CFU/mL of both the WT and mutant strain were determined over a time-course of three 12 hr passages (*Figure 4G*). When compared to the WT, a Δ*sodA::aph* mutant was less fit in both urine and LB, but the phenotype was exacerbated in urine. In contrast, the fitness of the remaining mutants was comparable to the parent strain when grown in LB medium and only less fit when grown in urine. While the relative competitive index of the Δ*ytfL::aph* strain was 0.8 in urine, the *ompA*, *fepB*, *fepD*, *exbB*, and *exbD* mutants were dramatically less fit than the parent strain in urine (*Figure 4G*).

Based on these observations, we hypothesized that iron levels were limiting growth in human urine and that the ability to acquire iron was crucial for *K. pneumoniae* to survive and grow in vivo (*Cassat and Skaar, 2013*; *Banerjee et al., 2020*). To test this hypothesis, urine was supplemented with exogenous ferrous iron (100 µM $FeSO_4$) or an iron chelator (100 µM bipyridine/2,2-dipyridyl). No variance in optical density was observed after passaging in LB medium (*Figure 5A*). However, the optical density of *K. pneumoniae* cultures decreased after passage in urine (*Figure 5B*), and the restricted growth was further exacerbated by the addition of the iron chelator 2,2-dipyridyl (*Figure 5C*). In contrast, supplementation of urine with exogenous iron increased growth relative to non-supplemented cultures with no difference in growth upon passage (*Figure 5D*). To further interrogate the role of iron limitation, the growth kinetics of the constructed mutants were measured in urine with and without exogenous ferrous iron for 16 hr in a microtiter plate assay. Not unexpectedly, area under the curve analysis revealed that mutants lacking genes involved in iron uptake (*fepB*, *fepD*, *exbB*, *exbD*) were drastically less fit than the WT (*Figure 6E*). Surprisingly, our screen showed several siderophore synthesis genes

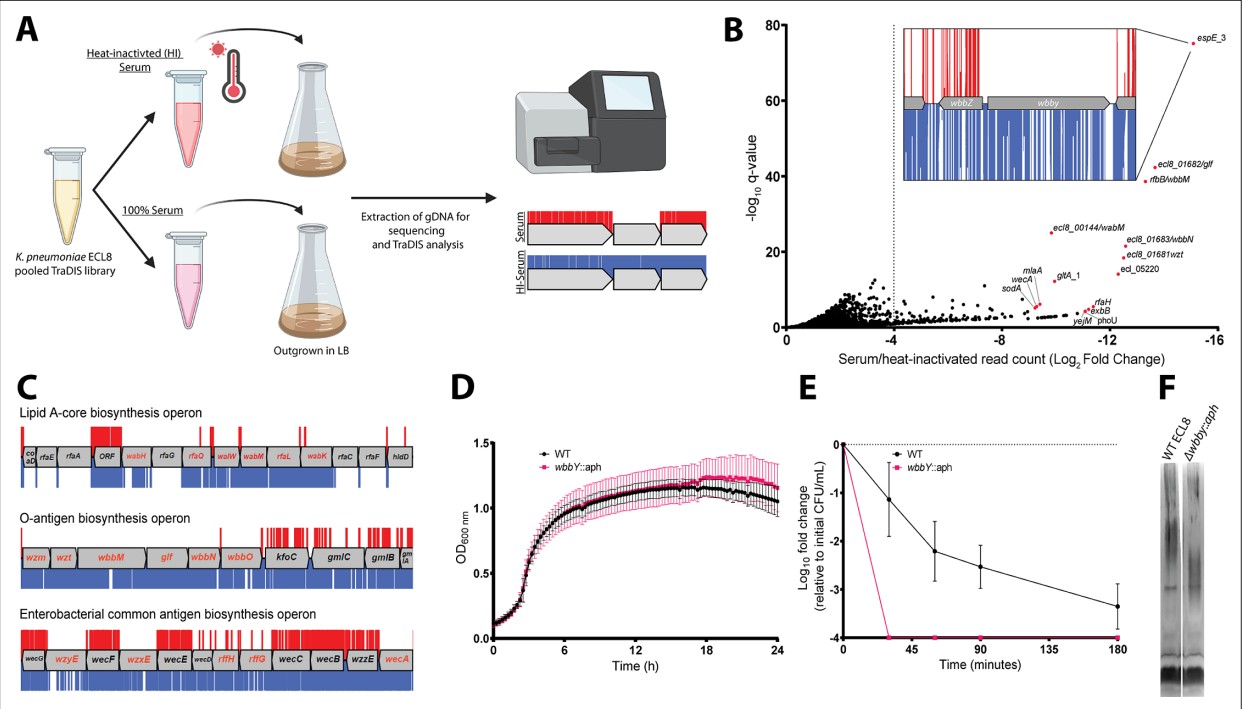

**Figure 6.** Overview and validation of ECL8 genes that increase resistance to complement-mediated killing. (**A**) The experimental methodology utilized for screening the transposon directed insertion-site sequencing (TraDIS) library in human serum and a heat-inactivated serum control. *K. pneumoniae* ECL8 ($2\times10^8$ cells) of the mutant library was inoculated into either 1 mL of human serum or 1 mL of heat-inactivated human serum and incubated for 90 min. Following exposure to serum, cells were grown to an $OD_{600}$ of 1 in Luria-Bertani (LB) medium to enrich for viable mutants. A 1 mL sample normalized to an $OD_{600}$ of 1 from each culture was processed for genome extraction and multiplexed sequencing using an Illumina MiSeq. (**B**) $Log_2FC$ for each gene of the *K. pneumoniae* ECL8 TraDIS library when incubated in pooled human serum relative to a heat-inactivated serum control. Selected genes highlighted in red are among the total of 144 genes that satisfy a stringent applied threshold ($log_2FC\geq-4$, Q-value$\leq0.05$). For brevity, only genes with a $log_2FC\geq0$ are illustrated. Inset: transposon insertion profile of *wbbY*, gene with the highest fold $log_2FC$, flanked by *wbbZ* and a transposable element at its 3'. Transposon insertions following exposure to serum and a heat-inactivated serum control are illustrated in red and blue, respectively. Transposon reads have been capped at a maximum of 10. (**C**) Transposon insertion profiles of genes within the: lipopolysaccharide (LPS), O-antigen, and the enterobacterial common antigen (ECA) biosynthesis operons. Genes in red font had a significantly ($log_2FC\geq-4$, Q-value$\leq0.05$) decreased fitness when disrupted with a transposon following exposure to serum for 90 min (red), relative to a heat-inactivated serum control (blue). Operons are not drawn to scale and reads capped at a maximum read depth of 1. (**D**) Growth profile of wild-type (WT) *K. pneumoniae* ECL8 and Δ*wbby::aph* in LB broth. Mean (*n*=3) is plotted (±1 SD). (**E**) Serum killing assay of WT *K. pneumoniae* ECL8 and Δ*wbby::aph*. Mean is plotted (±1 SD). (**F**) LPS profiles of WT *K. pneumoniae* ECL8 and Δ*wbby::aph*. Overnight cultures of each strain were normalized to an $OD_{600}$ of 1. The LPS was separated on 4–12% Bis-Tris gels and was visualized by silver staining using the SilverQuest kit (Invitrogen).

The online version of this article includes the following source data and figure supplement(s) for figure 6:

**Source data 1.** Essential gene table ECL8 (serum).

**Figure supplement 1.** Serum killing assay of *K. pneumoniae* ECL8 and *E. coli* BW25113.

**Figure supplement 2.** The Pearson correlation coefficient ($R^2$) of gene insertion index scores (IIS) for two sequenced biological replicates of the *K. pneumoniae* ECL8 transposon directed insertion-site sequencing (TraDIS) library following 90 min exposure to (blue) heat-inactivated serum or (red) serum.

did not confer a statistically significant fitness advantage for growth in urine (i.e. *entACDF*, *Figure 5F*). In contrast, the highest $log_2FC$ increase (~7-fold) in read counts mapped to the <u>F</u>erric <u>U</u>ptake <u>R</u>egulator (*fur*) gene indicating its loss is beneficial for growth in urine. In Gram-negative bacteria, Fur is a key regulator for iron homeostasis functioning as a transcriptional repressor for siderophore synthesis genes in an iron concentration-dependent manner (*Huang et al., 2012*; *Troxell and Hassan, 2013*; *Seo et al., 2014*). Furthermore, the Fur regulon includes the cognate ferric enterobactin uptake (Fep) genes, for which decreased $log_2FC$ read counts were also noted (*Figure 5G*). Altogether, this suggests that the Fur regulon plays a pivotal role for *K. pneumoniae* growth in urine by tightly regulating enterobactin synthesis and enterobactin-mediated iron uptake.

# Identification of genes required for resistance to complement-mediated killing

To identify genes that protect against complement-mediated killing, previously defined as the serum resistome (**Short et al., 2020**; **Phan et al., 2013**), we first established the serum killing kinetics of *K. pneumoniae* ECL8. Approximately, $2\times10^8$ cells were incubated with 100 µL of human serum for 180 min. At specified time points, aliquots were plated to determine the number of viable bacteria. As expected, no viable bacteria were recovered from the *E. coli* BW25113 control (**Doorduijn et al., 2021**). In contrast, a 2 $\log_{10}$ decrease in the number of viable *K. pneumoniae* cells was observed after 90 min exposure to normal human serum, and a >4 $\log_{10}$ decrease after 180 min exposure. To determine whether the observed reduction in viable cells was due to the presence of active complement proteins, *K. pneumoniae* was incubated in heat-inactivated serum control where no reduction in viable was noted (**Figure 6—figure supplement 1**).

Subsequently, $2\times10^8$ cells from the ECL8 TraDIS library were inoculated into 1 mL of human serum or 1 mL of heat-inactivated human serum and incubated for 90 min. Following exposure to serum, cells were grown in duplicate to an $OD_{600}$ of 1 in LB medium to enrich for viable mutants. DNA was harvested from each sample, and subsequently sequenced as described previously (**Figure 6A**). The Pearson correlation coefficient ($R^2$) of gene IIS for two sequenced biological replicates were calculated as 0.91 (normal human serum) and 0.76 (heat-inactivated serum) (**Figure 6—figure supplement 2**). To ensure robust data analysis, thresholds were applied to the data where genes with less than 50 mapped reads in the input library were removed from downstream analysis to exclude confounding essential genes and minimize the effect of stochastic mutant loss. A total of 356 genes with a $\log_2FC\leq-2$, $p<0.05$, and Q-value of ≤0.05 between the serum and heat-inactivated control were classified as genes that are advantageous for growth under complement-mediated killing conditions (**Figure 6B**). The complete list of all genes analyzed for growth in human serum and their associated statistical significance metrics are listed in **Figure 6—source data 1**. The operons encoding LPS core antigen, O-antigen, and enterobacterial common antigen biosynthesis were noted as containing several conditionally essential genes (**Figure 6C**). Mutants defective in these cell envelope structures have previously been reported to play a role in membrane integrity and resistance to serum killing for a variety of Gram-negative bacteria (**Onsare et al., 2015**; **Salamon et al., 2020**; **Rai and Mitchell, 2020**). Genes implicated as important for serum resistance encode proteins with an important role in membrane stability. They include GalE involved in LPS sugar subunit biosynthesis (**Frirdich and Whitfield, 2005**; **Ramos et al., 2012**), outer membrane protein MlaA involved in phospholipid retrograde transport (**Yeow et al., 2018**), cardiolipin synthase ClsA (**Tan et al., 2012**), Bam complex component BamE (**Rigel et al., 2012**), and proteins involved in LPS biosynthesis (LpxC) and its regulator (YejM) (**Nguyen et al., 2020**).

To validate our observations, we selected the gene with the greatest $\log_2FC$ decrease between serum and a heat-inactivated control, a putative glycosyltransferase *espE_3/wbbY* (inset, **Figure 6B**). Analyses using the 'Kaptive' webtool suggested that ECL8 encodes the O-antigen serotype O1v2 with the operon having 98.98% nucleotide identity to the reference genome (**Figure 7—source data 1**; **Lam et al., 2022**). The O-antigen of the O1v2 serotype features distal repeating D-galactan I and D-galactan II sugar subunits (**Hsieh et al., 2014**). Despite its role in O-antigen biosynthesis, *wbbY* is not located within the *rfb* LPS operon. To determine the role of *wbbY* in serum sensitivity, the gene was replaced with an *aph* cassette, and the resulting mutant was compared to its parent strain using a serum bactericidal assay. A comparable growth profile was observed for *K. pneumoniae* ECL8 and the Δ*wbbY::aph* mutant in LB medium (**Figure 6D**). However, the Δ*wbbY::aph* mutant was more sensitive to human serum than the parent with no viable mutants surviving after 30 min (**Figure 6E**). To investigate whether the *wbbY* deletion affected LPS production, crude LPS extracts were visualized using silver staining (**Figure 6F**). The Δ*wbbY::aph* strain produced LPS with a lower molecular weight compared to the parent strain, suggesting that a truncated O-antigen was expressed. Alterations in the amount or length of LPS produced are known to influence susceptibility to serum-mediated killing (**Salamon et al., 2020**). Importantly, Hsieh et al. have previously reported that deletion by *wbbY* in *K. pneumoniae* NTUH-K2044 (ST23) resulted in LPS profile defects due to abrogated D-galactan II production (**Hsieh et al., 2014**). The authors also demonstrate that gene complementation restores LPS profiles and serum resistance, thus demonstrating that this screen was robust in identifying a key *K. pneumoniae* gene involved in complement-mediated killing.

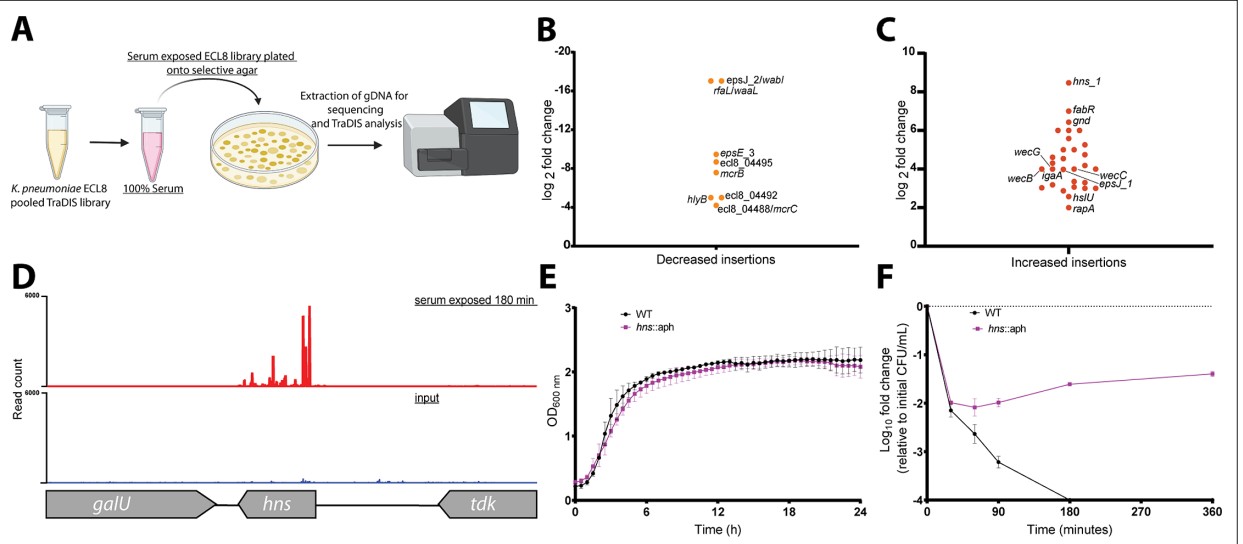

**Figure 7.** Overview and validation of ECL8 fitness factors for survival in pooled human serum. (**A**) The experimental methodology utilized for screening the transposon directed insertion-site sequencing (TraDIS) library to identify genetic factors that increase resistance to human serum. *K. pneumoniae* ECL8 ($2\times10^8$ cells) of the mutant library was inoculated into 1 mL of human serum and incubated for 180 min and compared to before serum exposure input control. The output pool was washed with PBS and plated onto Luria-Bertani (LB) agar supplemented with kanamycin. Following overnight growth, ~150,000 colonies were recovered and pooled for sequencing. Comparative analysis using AlbaTraDIS software depicting genes with (**B**) decreased insertions suggesting a loss of fitness or (**C**) increased insertions suggesting a gain of fitness to serum exposure. (**D**) Transposon and read count insertion profiles of *hns* locus: red illustrating pooled mutant serum exposed for 180 min and blue denoting the before serum exposure input control. (**E**) Growth profile of wild-type (WT) *K. pneumoniae* ECL8 and Δ*hns::aph* in LB broth. Mean (*n=3*) is plotted (±1 SD), where n=3. (**F**) Serum killing assay of WT *K. pneumoniae ECL8* and Δ*hns::aph*. Mean is plotted (±1 SD), where n=3.

The online version of this article includes the following source data and figure supplement(s) for figure 7:

**Source data 1.** Kaptive webserver results – ECL8 O-antigen K-antigen.

**Source data 2.** AlbaTraDIS ECL8 180 serum exposure results.

**Figure supplement 1.** Pearson correlation coefficient ($R^2$) of two biological replicates from the output transposon directed insertion-site sequencing (TraDIS) library following exposure to human serum for 180 min.

**Figure supplement 2.** Serum killing assay of wild-type (WT) *K. pneumoniae* ECL8, hns7::Tn5, and hns18::Tn5.

**Figure supplement 3.** TraDIS insertion profiles for mutant conferring increased resistance to serum killing.

## Identification of genes required for serum resistance

Following the identification of genes that when mutated confer increased sensitivity to complement-mediated killing (i.e. the serum resistome), the *K. pneumoniae* ECL8 TraDIS library was screened for mutants that confer increased resistance to serum killing. Approximately $2\times10^8$ TraDIS mutants were incubated in 1 mL of human serum for 180 min. The output pool was washed with PBS and plated onto LB agar supplemented with kanamycin (***Figure 7A***). Following overnight growth, ~150,000 colonies were recovered and pooled for sequencing confirming good correlation of replicates (***Figure 7—figure supplement 1***). We then used AlbaTraDIS to identify mutations that conferred a fitness advantage or disadvantage for survival in serum. In addition to identifying genes with an increase or decrease in reads between the condition and control samples, AlbaTraDIS will also analyze changes in sequencing read depth of insertions immediately upstream and downstream of a gene, as well as within-gene transposon orientation biases. Such differences can be strong indicators of mutations that result in gene expression or regulatory effects (for a detailed explanation, please see ***Page et al., 2020***). The data for insertions with a significant (>2-fold) change in read depth following serum exposure for 180 min are listed in ***Figure 7—source data 2***. As expected, similar to our previous experiment genes with the greatest significance in decreased insertions, suggesting decreased fitness, were O-antigen outer-core biosynthesis gene *espJ_2/wabI* and O-antigen ligase *rfaL/waaL* (***Figure 7B***). Genes with significantly increased insertions, suggesting increased fitness, included *hns*, *gnd*, and *igaA* (***Figure 7C***). Closer inspection of the transposon insertion profiles further confirmed the

enrichment of *hns* mutants when comparing the serum-exposed test to the input control (**Figure 7D**). To validate our observations, a defined *hns* mutant was constructed by allelic replacement with an *aph* cassette in the same orientation as native gene transcription. A comparable growth profile in LB was observed for the *hns* mutant relative to the parent (**Figure 7E**). To determine whether the *hns* mutant was indeed more resistant to serum a sample of $1 \times 10^8$ cells was incubated in serum for 360 min. At 30 min post-incubation the number of viable *hns*::aph cells was decreased by ~2 log. However, no additional decrease in viable cell numbers was observed at later time points and these strains proliferated in serum (**Figure 7F**). We also attempted to complement this phenotype but were unsuccessful, presumably due to the pleiotropic nature of *hns* mutants (**Figure 7—figure supplement 2**). Ares and colleagues demonstrated that *rcsA; galF; wzi;* and *manC* were derepressed in a Δ*hns K. pneumoniae* mutant resulting in increased capsule production (**Ares et al., 2016**). Similarly, upregulation of *rcsA* was noted in an *E. coli* Δ*hns* mutant. RcsA is known to positively regulate the capsular locus (**Sledjeski and Gottesman, 1995**; **Ebel and Trempy, 1999**). Notably, IgaA represses the Rcs regulon, preventing activation of RcsA. Mutants in *igaA* are enriched in our experiments (**Figure 7—figure supplement 3A**). These data suggest that activation of the Rcs regulon confers resistance to serum killing in *K. pneumoniae*, as noted for other members of the Enterobacteriaceae, by increasing capsule production (**Meng et al., 2021**).

The *gnd* gene encodes a 6-phosphogluconate dehydrogenase involved in the oxidative branch of the pentose phosphate pathway. Interestingly, sequence reads mapping to *gnd* were enriched for the forward orientation of the transposon only (**Figure 7—figure supplement 3B**). Transposon insertions in this orientation have transcriptional read-through driving expression of the downstream neighboring gene *manC* and more distal O-antigen biosynthesis genes (i.e. *rfb* cluster) (**Follador et al., 2016**). The *manC* gene encodes mannose-1-phosphate guanylyltransferase that is required for the synthesis of capsule, and as noted above, upregulation of *manC* results in increased production of capsule.

## Conclusion

For a variety of reasons, the determination of gene essentiality from TraDIS data is challenging (**Goodall et al., 2018**). Here, we used two different computational approaches to identify essential genes ensuring that our essential gene dataset includes those genes that can tolerate insertions in regions of the gene but where specific domains are essential for growth. Coupled with the presented phylogenetic and genomic analyses, the observation that 57% of the essential genes identified in this study were found to be essential in three of the four *K. pneumoniae* TraDIS libraries described to date provides reassurance that this library would provide meaningful data for subsequent studies. A minor caveat that is worthwhile mentioning is the potential for false-positive gene essentiality calls due to nucleoid-associated proteins hinders transposon insertion (**Choe et al., 2023**). Nevertheless, the high density of insertion sites achieved in our study ensured reliable identification of essential and conditionally essential genes and a low probability of false-positive identification, which had previously been noted for low complexity libraries (**Chao et al., 2016**).

The search for genetic determinants of bacterial survival and growth in vivo has been augmented by whole-genome approaches to infection, including transposon insertion site sequencing techniques such as TraDIS and TnSeq (**Cain et al., 2020**). These data have potential implications for the design and development of novel drugs, vaccines, and antivirulence strategies. In this study, we employed TraDIS to identify genes which, when mutated, confer decreased or increased fitness for growth in human urine and serum, two in vivo niches relevant to *K. pneumoniae* infection. Unexpectedly, our study revealed only 11 genes significantly required for growth in urine. These included genes encoding the outer membrane protein OmpA, its chaperone Skp, the superoxide dismutase SodA, and genes required for iron acquisition by the enterobactin uptake system. Notably, not all the genes for enterobactin synthesis, export, and uptake were essential (**Figure 4**). The gene encoding FepB, the periplasmic chaperone for the iron-laden siderophore, was essential, as were *fepD* and *fepG*, which encode the inner membrane uptake system. The TonB complex (TonB-ExbB-ExbD), which energizes the translocation of iron-laden enterobactin through the FepA outer membrane pore, was also essential for growth in urine. In contrast, the genes encoding FepA, the inner membrane ATPase FepC, and the enterobactin synthesis (Ent) and export (TolC) components were not essential. These observations can be explained in two ways. First, as *K. pneumoniae* ECL8 contains paralogous copies

of *fepA* (*fepA*_1 and *fepA*_2) and *fepC* (*fepC*_1 and *fepC*_2), the non-essential nature of *fepA* and *fepC* might be the result of functional redundancy. Second, as TraDIS is in principle a large-scale competition experiment between thousands of mutants, mutants lacking enterobactin (or aerobactin) synthesis and export genes can 'cheat' by acquiring the siderophore released from siderophore-producing strains within the population, a phenomenon described previously (*Butaitè et al., 2017*; *Holden and Bachman, 2015*). It should be mentioned, our study utilized filter sterilized urine which may preclude identification of genes that are required for interspecies predation, cross-protection, and cross-feeding in the context of a urinary microbiome (*Mataigne et al., 2021*; *Goh et al., 2023*).

In contrast to the limited number of genes in the urinome, the serum resistome consisted of more than 144 genes. This number is significantly greater than reports of a recent study that defined the serum resistome of four independent *K. pneumoniae* strains (*Short et al., 2020*). While that study found 93 genes that were required for survival in one or more strains, remarkably only three genes were common to all four strains: *rfaH*, *lpp*, and *arnD*. Comparison with our dataset revealed that in contrast to *rfaH*, *arnD* was not required for serum resistance in *K. pneumoniae* ECL8. Surprisingly, comparison of our data with the 56 genes comprising the serum resistome of the more distantly related multidrug-resistant *E. coli* strain EC958 revealed 14 genes in common. The variation in the outcome of these experiments can be accounted for in different ways. First, in all cases the genes encoding O-antigen and/or capsule synthesis were required for growth in normal human serum. As the genes encoding these surface structures vary with serotype, they are not identified as common genes, though ostensibly they have the same function. Second, the methodology appears at first glance to be identical in each study (growth in the presence of serum for 90 min). However, not all strains exhibit the same killing kinetics in normal human serum. Using fixed time points, strains with longer killing kinetics are likely to have fewer genes identified in TraDIS experiments, whereas strains with short killing windows will have more. Third, several of the genes identified in the serum resistome of other strains were excluded from our downstream analysis as there were fewer than 50 transposon insertions in the gene following exposure to heat-inactivated serum. A case in point is the *lpp* gene. This observation indicates that *lpp* is required for *K. pneumoniae* ECL8 growth in human serum but not resistance to complement-dependent killing per se. Fourth, the density of the library used, the level of comparative annotation, and the thresholds to define significance, all have the potential to influence the final outcome of such experiments. Finally, strain-specific features, such as gene duplication events, functional redundancy in biosynthetic pathways, and the variation in the genetic complement of each strain, are likely to influence the contribution each gene makes to serum resistance.

Despite the intra- and interspecies variation in the serum resistome noted above, comparison of the urinome and serum resistome of *K. pneumoniae* ECL8 revealed that mutation of five genes (*ompA*, *galE*, *exbB*, *exbD*, and *sodA*) conferred reduced fitness for both growth in urine and survival in serum. The role of *exbB* and *exbD* is described above but their detection in both experimental conditions reinforces the importance of iron acquisition for infection. The major outer membrane protein OmpA identified in this study encodes a functionally pleiotropic outer membrane protein (*Huang et al., 2012*; *Troxell and Hassan, 2013*). Notably, a role for OmpA in serum resistance has previously been described for *E. coli*, however it was not identified by TraDIS experiments investigating the serum resistome (*Phan et al., 2013*). The molecular basis for this discrepancy remains unresolved. GalE is known to play a key role in serum resistance for *E. coli* and *Salmonella* spp. The gene *galE* encodes a UDP-glucose 4-epimerase catalyzing the interconversion of UDP-D-galactose/UDP-D-glucose (*Frirdich and Whitfield, 2005*). It has been previously reported that GalE plays a role in D-galactan II synthesis, an O1 LPS component in *K. pneumoniae* (*Ramos et al., 2012*; *Clarke and Whitfield, 1992*). These data suggest that loss of *galE* results in a modified LPS in *K. pneumoniae* resulting in increased serum susceptibility. Additionally, the manganese-dependent superoxide dismutase SodA converts superoxide to hydrogen peroxide and oxygen to prevent DNA and other cellular damage (*Touati, 1983*; *Doukyu and Taguchi, 2021*). A previous study in *K. pneumoniae* has reported the inducible expression of *sodA* in a triumvirate system with *sodB* and *sodC*, expressed in response to anaerobic challenge and high-oxygen concentration (*Najmuldeen et al., 2019*). These transcriptional responses might very well be present in both the microenvironments of blood vessels and the bladder, in the form of nutrient limitation and oxygen exposure (*Flores-Mireles et al., 2015*; *Murdoch and Skaar, 2022*; *Eberly et al., 2017*). Hence, the observed importance of *sodA* might be a consequence of elevated levels of reactive oxygen species due to iron acquisition and could represent a specialized

metabolic adaptation for growth in these environments. Ultimately, these findings underline the interplay between bacterial iron homeostasis and oxidative stress previously reported in Gram-negative bacteria (*Cornelis et al., 2011*; *Adler et al., 2014*; *Peralta et al., 2016*; *Guest et al., 2019*; *Singh et al., 2022*). However, these data shine a light on cross-niche (i.e. urine and serum) genes and associated pathways as suitable drug development targets to treat *Klebsiella* infections. Finally, this library was then used to identify genes conferring serum resistance. An interesting finding was the observed transposon insertion orientation bias into *gnd*. As the downstream transcriptional unit of *gnd* are O-antigen-related genes (i.e. biosynthesis and export), this in turn lends further credence to a transcriptional regulatory model where biosynthesis of O-antigen and capsule is coupled for biophysical reasons as has been previously reported (*Dorman et al., 2018*; *Singh et al., 2022*). Collectively, these data show that O-antigen biosynthesis genes (i.e. *wabl* and *waaL*) play a contributing role in serum resistance while *hns* mutants were shown to be more resistant to serum killing than WT *K. pneumoniae* ECL8. However, the mechanism that underpins this resistance is still largely unknown and requires further characterization. Despite the previously described links between *hns* and the regulation of a selection of capsular genes (*Dorman et al., 2018*; *Ares et al., 2016*), this is the first report of a role for *hns* in resisting complement-mediated killing in human serum.

This genome-wide screen has identified a suite of genetic fitness factors required for growth of *K. pneumoniae* ECL8 in human urine and serum. The defined ECL8 urinome and serum resistome gene sets from this study differ from other TraDIS studies, hinting that the genotype of different pathogenic lineages might only be a piece in understanding the underlying infection biology. Future studies like these but expanding to other isolates, and the lesser studied clinically relevant *K. pneumoniae* species complex, will be imperative for the effective future development of targeted therapeutics toward this problematic pathogen.

## Materials and methods
### Bacterial strains, plasmids, and culturing conditions
A complete list of bacterial strains, plasmids, and primers utilized in this study are listed in *Supplementary file 1A* and *Supplementary file 1B*. To culture strains, scrapings from a frozen glycerol stock strain were plated onto solid LB (10 g tryptone, 5 g yeast extract, 10 g NaCl) supplemented with agar 1.5% (wt/vol) and were incubated overnight at 37°C. A single colony was isolated and used to inoculate 5 mL of liquid LB medium incubated overnight at 37°C with 180 RPM shaking. To prepare stocks, bacteria were grown to mid-exponential phase in LB medium and stored at –80°C with 25% (vol/vol) glycerol. When required, solid or liquid media were supplemented with appropriate antibiotics at the following concentrations: ampicillin (35 µg/mL), chloramphenicol (100 µg/mL), and kanamycin (100 µg/mL).

### Generation of the *K. pneumoniae* ECL8 transposon mutant library
An overnight culture of *K. pneumoniae* ECL8 was inoculated into 2× YT broth at $OD_{600}$ of 0.05 supplemented with a final concentration of 0.7 mM EDTA and incubated at 37°C with 180 RPM shaking. Cells were harvested at 0.4 $OD_{600}$ by centrifugation (4000 × *g*) at 4°C for 20 min. Cell pellets were washed four times with ice-cold 10% (vol/vol) glycerol. *K. pneumoniae* ECL8 electrocompetent cells were transformed with 0.2 µL of EZ-Tn*5* transposon (Epicentre) at 1.4 kV and recovered in brain heart infusion broth for 2 hr at 37°C with 180 RPM shaking. Cells were plated onto LB agar supplemented with 50 µg/mL kanamycin and incubated overnight at 37°C. More than 1 million transformants were pooled in 15% (vol/vol) LB glycerol for storage at –80°C until required.

### Transposon library screening in human urine
Human urine from seven healthy male volunteers was sterilized using a vacuum filter (0.2 µM). Urine was pooled and stored at –80°C until required. Approximately $2×10^8$ *K. pneumoniae* ECL8 TraDIS mutants were inoculated into 50 mL of urine and grown for 12 hr at 37°C with shaking (P1) in biological duplicates in parallel with control samples that were instead inoculated into 50 mL of LB medium. A sample of P1 was inoculated into 50 mL of urine or LB medium, respectively, at an initial $OD_{600}$ of 0.05 and grown for a subsequent 12 hr (P2). Both control and test experiments were passaged and

grown for a further 12 hr (P3). A sample of culture from P3 was removed and normalized to an OD$_{600}$ of 1 for subsequent genomic DNA extraction.

## Transposon library screening in human serum

A 10 mL sample of human blood was collected from eight healthy volunteers (male and female). Blood was pooled and centrifuged at 6000 × $g$ for 20 min to separate sera from blood components. The pooled serum was sterilized, aliquoted, and stored at –80°C until required. Approximately 2×10$^8$ *K. pneumoniae* ECL8 mutants were inoculated into 1 mL of pre-warmed sera, heat-inactivated sera (60°C for 1 hr) in biological duplicates. Samples were incubated at 37°C for 90 min with 180 RPM shaking. Following this, cells were harvested by centrifugation at 6000 × $g$ for 10 min and washed twice with PBS. Cells were then inoculated into 50 mL of LB for outgrowth at 37°C with 180 RPM shaking and harvested at an OD$_{600}$ of 1 by centrifugation at 6000 × $g$ for 10 min. A 1 mL sample of each culture was used for genomic DNA extraction.

## Transposon library sequencing and data analysis

Pooled kanamycin-resistant *K. pneumoniae* ECL8 cells were prepared for sequencing following an amended TraDIS protocol (*Goodall et al., 2018*). Briefly, genomic DNA from the *K. pneumoniae* ECL8 library of pooled mutants was extracted from ~1×10$^9$ cells in biological duplicate, as per the manufacturer's instructions (QIAGEN QIAamp DNA blood minikit). The DNA concentration of replicates was determined using the Qubit 2.0 fluorometer (Thermo Fisher Scientific). The DNA was sheared into ~200 bp fragments by ultra-sonication (Bioruptor Plus Diagenode). Fragmented DNA samples were prepared for Illumina sequencing using the NEBNext Ultra I DNA Library Prep Kit for Illumina according to the manufacturer's instructions with the following modifications: ends of fragmented DNA were repaired using the End Prep Enzyme Mix (NEB) and an NEBNext adaptor for Illumina sequencing was ligated to the newly repaired ends. During protocol optimization, samples were analyzed using a Tapestation 2200 (High Sensitivity D5000) to determine DNA fragment sizes following adapter ligation. The uracil within the adaptor hairpin loop was enzymically excised using the USER enzyme (NEB). AMPure XP SPRI beads (Beckman Coulter) or custom-made SPRI beads were used to select for DNA fragments ~250 bp in size.

To enrich for fragments containing the transposon a custom PCR step was introduced using a forward primer annealing to the 3' end of the antibiotic marker of the transposon and a reverse primer that annealed to the ligated adaptor. The PCR product was purified using SPRI beads at a ratio of 0.9:1 (beads:sample). A further custom PCR step introduced sequencing flow-cell adaptors (Illumina barcodes) and inline barcodes to allow for sample multiplexing (*Supplementary file 1C*). Samples were then stored at –20°C until sequencing. After qRT-PCR (KAPA Library Quantification Kit Illumina Platforms) to determine TraDIS library concentration, a 1.5 µL sample of each library sample diluted to 8 nM was combined and mixed by pipetting to generate a pooled amplified library (PAL) of samples for multiplexed sequencing. The PAL was prepared for Illumina MiSeq sequencing according to the manufacturer's instructions.

Bioinformatic analysis of sequencing reads was completed on the Cloud Infrastructure for Microbial Bioinformatics (CLIMB) (*Connor et al., 2016*). Raw fastQ files were first demultiplexed according to Illumina barcodes automatically using the Illumina MiSeq software. Experimental replicates were separated by custom inline barcodes using the FastX barcode splitter and the barcodes were subsequently removed using the FastX trimmer (v0.0.13). Each read was checked for the presence of the transposon sequence in two steps: first, reads that contained the first 25 bp of the transposon sequence (AGCT TCAGGGTTGAGATGTGTA), introduced by the TKK_F primer, allowing for three mismatches were filtered and parsed. Parsed reads were subsequently checked for the presence of the final 10 bp of the transposon sequence (TAAGAGACAG), allowing for one mismatch. The transposon sequence and reads <20 bp were trimmed using Trimmomatic (v0.39). Resulting reads were mapped to the *K. pneumoniae* ECL8 (HF536482.1 and HF536483) genome and plasmid using the Burrows-Wheeler Alignment tool (BWA-MEM) using default parameters (i.e. -k=20 bp exact match). Mapped reads were indexed using samtools (v1.8) and converted into bed format using the Bam2bed tool (Bedtools suite v2.27.1). The resulting bed file was intersected against the annotated CDS in the *K. pneumoniae* ECL8 gff file, which was generated using PROKKA (v1.14.0). Transposon insertion sites and their

corresponding location were quantified using custom Python scripts. Mapped transposon insertion sites were visualized using the Artemis genome browser (*Rutherford et al., 2000*).

## Statistical analysis of transposon insertion density

A modified geometric model was applied to identify the probability of finding k-1 insertion-free bases followed by a transposon insertion as reported previously (*Goodall et al., 2018*). In a string of 10,000 independent trials the probability of an insertion 'p' was calculated using the following equation: P(k)=p(1–p)(k).

## Identification of putative essential genes

In-house scripts kindly provided by *Langridge et al., 2009*, were amended and used for the prediction of essential genes. Briefly, the number of unique transposon insertion sites for each gene was normalized for CDS length (number of UIPs/CDS length in bp), which was denoted the IIS. The Freedman-Diaconis method was used to generate a histogram of the IIS for each gene of the *K. pneumoniae* ECL8 TraDIS library. Distributions were fitted to the histogram using the R MASS library (v4.0.0).

An exponential distribution was applied to the 'essential' mode, situated on the left of the histogram. A gamma distribution was applied to the 'non-essential' mode, situated on the right of the histogram. The probability of a gene belonging to each mode was calculated and the ratio of these values was denoted as the log-likelihood score. Genes were classified as 'essential' if they were 12 times more likely to be situated in the left mode than the right mode. Genes with log-likelihood scores between the upper and lower $\log_2$(*Garbati and Al Godhair, 2013*) threshold values of 3.6 and –3.6 respectively were classified as 'unclear'. Genes with log-likelihood scores below the 12-fold threshold were classified as 'non-essential'.

## Genetic context and comparison of essential gene lists between ECL8, KPNIH1, RH201207, and ATC43816

Annotated genomes of *K. pneumoniae* KPNIH1 (Accession: CP009273.1), RH201207 (Accession: FR997879.1), and ATCC 43816 (Accession: CP009208.1) were downloaded from NCBI. Nucleotide sequences of putative essential genes were extracted using the SeqKit toolbox (*Shen et al., 2016*). *K. pneumoniae* ECL8 essential gene homologs were identified in KPNIH1, RH201207, and ATC43816 using Galaxy AU NCBI BLAST+ wrapper blastn using the following criteria (e-value>1e-10, percent identity>90, percent length>30) and utilizing the top hit based on calculated e-value and bitscore (*Cock et al., 2015*).

The phylogenetic and ANI analysis of ECL8 and other *K. pneumoniae* isolates (*Figure 3—source data 1*) was performed using the Integrated Prokaryotes Genome and Pan-genome Analysis (IPGA v1.09) webserver (*Liu et al., 2022*). Using the default settings of the genome analysis module, ANI values between each submitted genome pairs were calculated. Then, IPGA performed genome annotation based on entries in the gcType microbial genome database for the given target genome list for phylogenetic analysis and tree construction (*Shi et al., 2021*).

## Bio-TraDIS analysis for the identification of conditionally advantageous genes

To identify conditionally essential genes, we used the Bio-TraDIS analysis pipeline (*Barquist et al., 2016*), that measures the read count $\log_2$FC between each CDS. To ensure robust analysis, CDS with <50 sequence reads in either the test or control condition were filtered and binned. CDS with a $\log_2$FC >2 (test/control) and a Q-value of >0.05 were classified as conferring a fitness advantage under the conditions tested. To comparatively analyze the impact of transposon insertion events in genomic regions in the serum-resistant dataset (180 min) to the input, we used the AlbaTraDIS package (v1.0.1) with the following parameters: -a -c 10.0 -f 2 -m 0 (*Page et al., 2020*).

## Generation of knock-out strains

Mutants derived from *K. pneumoniae* ECL8 were constructed using the $\lambda$ Red recombinase system (*Datsenko and Wanner, 2000*). Briefly, the *K. pneumoniae* ECL8 strain was transformed with pACB-SCSE, which contains genes coding for the arabinose-induced $\lambda$ Red recombinase system that permits homologous recombination between dsDNA PCR products and target loci in the bacterial

genome. The recombination is based on short stretches of flanking homology arms (~65 bp) with the site of recombination. Using pKD4 as a template for the selectable kanamycin cassette, the PCR products used for replacing the target genes were amplified, gel extracted, and electroporated into electrocompetent strain *K. pneumoniae* ECL8 harboring pACBSCSE prepared in the presence of 0.5% (wt/vol) arabinose. Candidate mutants were screened using antibiotics, verified by PCR using primers outside the region of recombination then followed by Sanger sequencing.

## Bacterial growth assay

A single colony of each bacterial strain was inoculated into 5 mL of LB medium and grown overnight as previously described. Strains were normalized to an $OD_{600}$ of 1.00 (approximately $7\times10^8$ cells) and washed twice in PBS by centrifugation at $6000 \times g$ for 10 min. Cell pellets were resuspended in 1 mL of the growth medium required for the bacterial growth assay. Greiner Bio-One 96-well U-bottom microtiter plates were inoculated with bacterial strains at an initial $OD_{600}$ of 0.02 in a final well medium volume of 150 µL and sealed with a Breathe-Easy sealing membrane (Sigma-Aldrich). Inoculated plates were incubated at 37°C with 300 RPM shaking, $OD_{600}$ measurements were taken at 15 min over 24 hr using a CLARIOstar plate reader (BMG LABTECH).

## Urine co-culture competition studies between *K. pneumoniae* ECL8 and isogenic mutants

Overnight cultures of *K. pneumoniae* ECL8 and an isogenic mutant were normalized to an $OD_{600}$ of 1.00 and washed twice with PBS. Cell pellets were resuspended in 1 mL of urine. A 500 µL of either the WT or mutant culture were inoculated into 25 mL of urine and incubated at 37°C with 180 RPM shaking for 12 hr. The sample was then transferred to 25 mL of fresh urine at an initial $OD_{600}$ of 0.05 and grown for a further 12 hr. The culture was passaged for a further 12 hr of growth. These cultures were serially diluted in PBS at 0, 6, 12, 24, and 36 hr and plated onto LB agar. Following overnight growth, agar plates were replica plated onto either LB agar or LB agar supplemented with 100 µg/mL kanamycin grown overnight at 37°C then enumerated. The number of WT colony forming units (CFUs) was calculated by subtracting the number of CFUs in media with antibiotics from the total number of CFUs in media without antibiotics. Relative fitness was calculated as a competitive index, defined as ratio of mutant:WT viable cells divided by the corresponding inoculum.

## LPS extraction

A single colony of ECL8 was inoculated into 5 mL of LB medium and incubated overnight at 37°C with 180 RPM shaking. Overnight cultures were normalized to an $OD_{600}$ of 1.00 and centrifuged at $14,000 \times g$ for 10 min. Cells were resuspended in 100 µL of cracking buffer (0.125 M Tris-HCl, 4% [wt/vol] SDS, 20% [vol/vol] glycerol, 10% [vol/vol] 2-mercaptoethanol in $dH_2O$). The cell suspension was incubated at 100°C for 5 min and transferred to –80°C for 5 min. The cell suspension was incubated at 100°C for a further 5 min and centrifuged at $14,000 \times g$ for 10 min. An 80 µL sample of the supernatant was transferred to a new tube and incubated with 5 µL of 5 mg/mL Proteinase K (QIAGEN) for 1 hr at 60°C. Samples were diluted 2× with Laemmli sample buffer (Sigma-Aldrich) and incubated at 95°C for 5 min. Samples were loaded onto a 4–12% Bis-Tris precast NuPAGE gel (Invitrogen) and run at 150 V for 1.5 hr. LPS was visualized by silver staining using the SilverQuest kit (Invitrogen) following the manufacturer's instructions.

## Serum bactericidal assay

Pooled serum was thawed on ice and pre-warmed to 37°C. Overnight cultures of bacterial strains were normalized to an $OD_{600}$ of 1.00 in LB medium. Bacterial cells were washed twice with PBS and resuspended in a final volume of 1 mL PBS. A 50 µL sample of the cell suspension was inoculated into 50 µL of sera. Sera and cell samples were incubated at 37°C with 180 RPM shaking. Viable bacterial cells were enumerated by plating 10 µL of cell and sera suspension onto LB agar over a time-course: 0 min, 30 min, 60 min, 90 min, 180 min. Serum bactericidal activity was plotted as the $log_{10}$ change in CFU relative to the initial inoculum.

## Additional information

### Funding

| Funder | Grant reference number | Author |
|---|---|---|
| Biotechnology and Biological Sciences Research Council | MIBTP | Jessica Gray Emily Goodall |

The funders had no role in study design, data collection and interpretation, or the decision to submit the work for publication.

### Author contributions

Jessica Gray, Conceptualization, Data curation, Formal analysis, Validation, Investigation, Visualization, Methodology, Writing - original draft, Project administration, Writing - review and editing; Von Vergel L Torres, Data curation, Formal analysis, Validation, Investigation, Methodology, Writing - original draft, Project administration, Writing - review and editing; Emily Goodall, Conceptualization, Data curation, Formal analysis, Supervision, Validation, Investigation, Visualization, Writing - review and editing; Samantha A McKeand, Data curation, Formal analysis, Investigation, Visualization, Methodology, Writing - review and editing; Danielle Scales, Resources, Data curation, Formal analysis, Validation, Methodology, Writing - review and editing; Christy Collins, Data curation, Formal analysis, Funding acquisition, Validation, Visualization, Methodology; Laura Wetherall, Data curation, Software, Formal analysis, Validation, Investigation, Methodology; Zheng Jie Lian, Data curation, Validation, Investigation, Visualization, Methodology; Jack A Bryant, Data curation, Software, Formal analysis, Supervision, Funding acquisition, Validation, Investigation, Visualization, Methodology, Writing - original draft, Writing - review and editing; Matthew T Milner, Resources, Data curation, Software, Formal analysis, Supervision, Validation, Investigation, Visualization, Methodology, Writing - review and editing; Karl A Dunne, Conceptualization, Resources, Data curation, Software, Formal analysis, Supervision, Investigation, Methodology, Writing - original draft, Project administration; Christopher Icke, Conceptualization, Formal analysis, Supervision, Funding acquisition, Investigation, Visualization, Methodology, Writing - original draft, Writing - review and editing; Jessica L Rooke, Formal analysis, Supervision, Validation, Investigation, Visualization, Methodology, Writing - original draft, Project administration, Writing - review and editing; Thamarai Schneiders, Conceptualization, Resources, Methodology, Writing - original draft, Writing - review and editing; Peter A Lund, Data curation, Formal analysis, Supervision, Investigation, Methodology, Writing - original draft, Writing - review and editing; Adam F Cunningham, Conceptualization, Supervision, Funding acquisition, Methodology, Project administration, Writing - review and editing; Jeff A Cole, Conceptualization, Formal analysis, Supervision, Writing - original draft, Project administration, Writing - review and editing; Ian R Henderson, Conceptualization, Resources, Data curation, Software, Formal analysis, Supervision, Funding acquisition, Validation, Investigation, Visualization, Methodology, Writing - original draft, Project administration, Writing - review and editing

### Author ORCIDs

Von Vergel L Torres  https://orcid.org/0000-0003-3387-1112
Samantha A McKeand  https://orcid.org/0000-0001-5579-7422
Zheng Jie Lian  https://orcid.org/0000-0002-3054-1866
Christopher Icke  https://orcid.org/0000-0002-7815-8591
Ian R Henderson  https://orcid.org/0000-0002-9954-4977

Reviewer #1 (Public Review): https://doi.org/10.7554/eLife.88971.3.sa1
Reviewer #3 (Public Review): https://doi.org/10.7554/eLife.88971.3.sa2
Author response https://doi.org/10.7554/eLife.88971.3.sa3

## Additional files

### Supplementary files

• Supplementary file 1. Strains, plasmids, and primers used in this study. (A) Bacterial strains and plasmids utilized in this study. (B) Primer nucleotide sequences for construction of *K. pneumoniae* chromosomal mutant strains. (C) Primer nucleotide sequences for enrichment of the transposon junction (TKK_F and TKK_R) and the introduction of an inline barcode for multiplexed sequencing (TKK 6, 7, 8, 9).

• MDAR checklist

### Data availability

Sequencing data have been deposited in ENA under the accession code PRJEB76649.

The following dataset was generated:

| Author(s) | Year | Dataset title | Dataset URL | Database and Identifier |
|---|---|---|---|---|
| Goodall E, Gray J | 2024 | Transposon mutagenesis screen in Klebsiella pneumoniae | https://www.ebi.ac.uk/ena/browser/view/PRJEB76649 | European Nucleotide Archive, PRJEB76649 |

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
