## [Editor Report · eLife assessment]

This **valuable** study is of relevance for those interested in the mechanism required for infections of humans by *Klebsiella pneumoniae*. The authors apply TraDIS (high-density TnSeq) to *K. pneumoniae* with the goal of identifying genes required for survival under various infection-relevant conditions and the gene sets identified, together with the raw sequence data, will be resources for the Klebsiella research community. The evidence to support the lists of essential and conditionally-essential genes is **convincing**. The study provides strong evidence that some genes are conditionally essential in urine because of iron limitation, but there is less mechanistic insight for genes that are conditionally essential in serum.

---

## [Referee Report · Reviewer #1 (Public Review)]

The study provides strong evidence that some genes are conditionally essential in urine because of iron limitation.

The authors raise the intriguing possibility that some mutants can "cheat" by benefitting from the surrounding cells that are phenotypically wild-type. The authors make it clear that the proposed cheating mechanism is speculation, but there is a missed opportunity to test this hypothesis. I did not understand the authors' rationale for not doing this experiment.

In cases where there are disparities between studies, e.g., for genes inferred to be essential for serum resistance, it would be informative to test individual deletions for genes described as essential in only one study. The authors argue this is beyond the scope of the study. Their conclusions of the study are not impacted by the absence of these experiments, but readers will be left wondering which lists of conditionally essential genes are correct, or whether there are strain-dependent or condition-dependent contexts that influence conditional essentiality.

---

## [Referee Report · Reviewer #3 (Public Review)]

In this study Gray and coworkers use a transposon mutant library in order to define: (i) essential genes for K. pneumoniae growth in LB medium, (ii) genes required for growth in urine, (iii) genes required for resistance to serum and complement mediated killing. Although there are previous studies, using a similar strategy, to describe essential genes for K. pneumoniae growth and genes required for serum resistance, this is the first work to perform such a study in urine. This is important because these types of pathogens can cause urinary tract infections. Moreover, the authors performed the work using a highly saturated library of mutants, which makes the results more robust, and used a clinically relevant strain from a pathotype for which similar studies have not been performed yet. Besides applying the transposon mutant library coupled with high-throughput sequencing, the authors validate some of the most relevant genes required for each condition using targeted mutagenesis. This is an important step to confirm that the results obtained from the library are reliable. Although this was done for only a small subset of the most significant genes. In addition, in vitro experiments involving complementation of urine with iron provide additional support to the results obtained with the mutants suggesting the importance of genes required for iron acquisition in a limiting-iron environment such as urine. Overall, the study is well-designed and written, and the methodology and analysis performed are adequate. The study would have benefited from in vivo experiments, including a mouse model of bacterial sepsis or urinary tract infections which could have demonstrated the role of some of the identified genes in the infection process. Nevertheless, the results obtained are informative for the scientific community since they pinpoint genes potentially more relevant in infections caused by K. pneumoniae. The identified genes could represent future targets for developing new therapies against a type of pathogen that is acquiring resistance to all available antibiotics. Although, as mentioned above, these potential targets should be confirmed using in vivo models.

One potential weakness of the work is that the TnSeq analysis only included two replicates per condition, thus it is possible that some of the differences detected may not be reproducible in future studies, first of all those that are less significant. In this sense, hundreds of genes were detected to be theoretically relevant for bacterial resistance to complement in serum. It is possible that some of these genes represent false positives. Thus, confirmation of the relevance of these genes in resistance to complement should be performed in future studies.

---

## [Author Response]

The following is the authors’ response to the original reviews.

**Reviewer #1 (Public Review):**
(1) The data strongly suggest that iron depletion in urine leads to conditional essentiality of some genes. It would be informative to test the single gene deletions (Figure 3G) for growth in urine supplemented with iron, to determine how many of those genes support growth in urine due to iron limitation.

We appreciate this suggestion. We have now included this suggested experiment as a new panel (Figure 5G).

(2) Line 641. The authors raise the intriguing possibility that some mutants can "cheat" by benefitting from the surrounding cells that are phenotypically wild-type. Growing a fepA deletion strain in urine, either alone or mixed with wild-type cells, would address this question. Given that other mutants may be similarly "masked", it is important to know whether this phenomenon occurs.

We thank the reviewer for this suggestion but believe that this would be very difficult to ascertain in K. pneumoniae as several redundant iron uptake systems exist. This would require significantly more time to construct sequential/combinatorial iron-uptake mutants to exactly determine this “cheating” and “masking” phenomenon and such work is beyond the scope of the current study.

(3) In cases where there are disparities between studies, e.g., for genes inferred to be essential for serum resistance, it would be informative to test individual deletions for genes described as essential in only one study.

We thank the reviewer for this suggestion, and we agree that deleting conditionally essential genes (i.e. serum resistance) could help identify discrepancies in methodology with other studies but this is beyond the scope of this study. Furthermore, we do not have these other strains readily available to us and importing these strains into Australia is challenging due to the strict import/quarantine laws.

**Reviewer #1 (Recommendations For The Authors)**
(4) Line 529. Why was 50 chosen as the read count threshold?

This was chosen as the minimum threshold needed to exclude essential genes from the comparative analysis, as these can contribute false positive results where a change from, for example, 2 to 5 reads between conditions is considered a >2-fold change. We have updated the manuscript text to highlight this: “were removed from downstream analysis to exclude confounding essential genes and minimize the effect of stochastic mutant loss” line 539

(5) The titles for Figure 5 and Figure 6 appear to be switched.

Thank you, we have now corrected this error.

(6) Line 381. "Forty-six of these regions contain potential open reading frames that could encode proteins". How is a potential ORF defined?

This was based on submitting the selected 145bp regions to BLASTx using default parameters and listing the top hit (if one was found). We have now edited the manuscript text to make this clearer. (Line 394)

(7) Two previous TnSeq studies looking at *Escherichia coli* and Vibrio cholerae suggest that H-NS can prevent transposon insertion, leading to false positive essentiality calls. Is there any evidence of this phenomenon here? A/T content could be used as a proxy for H-NS occupancy.

We thank the reviewer for this point and also agree that H-NS or other DNA-binding proteins could indeed lead to false-positive essentiality calls using TraDIS. Based on this, we have now included a sentence in the conclusion section mentioning this methodological caveat (Line 631). We believe that A/T content could potentially be used as a proxy for H-NS occupancy,

**Reviewer #2 (Recommendations For The Authors):**
(1) The authors may wish to reformat the manuscript by decanting a number of panels and figures as supplementary material. These include the panels related to the description of TraDIS (for example Fig 1D, 1E, 1F. 1G, Fig 2A, Fig 3C, 3D, 3E, 3F, Fig 5C, Fig 6D). This is a well-established method.

We thank the reviewer for this suggestion but believe that these panels allow the methodology and resulting insertion plots to be more followable and allow other researchers, of varying expertise, to better understand this functional genetic screen technique.

(2) The authors need to indicate how relevant the strain they have probed is. Is it a good reference strain of the KpI group?

This is a great suggestion and we have now included a new figure illustrating the genetic context and relatedness of K. pneumoniae ECL8 within the KpI phylogroup (New Figure 3).

(3) The authors need to provide an extensive comparison between the data obtained and those reported testing other Klebsiella strains. A Table identifying the common and different genes, as well as a figure, may suffice. I would encourage authors to compare also their data against *E. coli* and *Salmonella*. For example, igaA seems to be not essential in Kebsiella although data indicates it is in *Salmonella*.

We thank the reviewer for their comment and appreciate that our data could be extended and compared to other relevant Enterobacteriaceae members. However, we believe this is beyond the scope of this study as the focus is more on K. pneumoniae.

(4) None of the mutants tested further are complemented. Without these experiments, it cannot be rigorously claimed that these loci play any role in the phenotypes investigated.

We agree that complementation is an important tenet for validation of mutant gene phenotypes to specific gene loci, in this case wbbY has already been complemented and believe complementation for an already known molecular mechanism would be redundant. Please refer to our response in point 6.

We complemented isolated transposon mutants hns7::Tn5 and hns18::Tn5 with a mid-copy IPTG inducible . We observed a slight increase in serum susceptibility but not full rescue of the WT phenotype (i.e. serum susceptibility). We suspect that the imperfect rescue of the serum-resistance phenotype observed could be due to the expression levels and copy number of the complement hns plasmid used. As hns is a known global regulator its possible pleiotropic role is complex as many aspects of stress response, metabolism or capsule could be affected in Klebsiella (doi.org/10.1186/1471-2180-6-72, doi.org/10.3389/fcimb.2016.00013). We have now included in the text our efforts in complementation and have included a new supplementary figure (Figure S11).

(5) The contribution of siderophores to survival in urine is not conclusively established. Authors may wish to test the transcription of relevant genes, and to assess whether the expression is fur dependent in urine. Also, authors may wish to identify the main siderophore needed for survival in urine by probing a number of mutants; this will allow us to assess whether there is a degree of selection and redundancy.

We thank the reviewer for their comment and agree siderophore uptake is important. We have now included an additional panel (Figure 5G) interrogating the importance of iron-uptake genes grown in urine which is iron limited. We do appreciate that further experiments looking into the Fur regulon and siderophore biosynthesis would be interesting but believe this is outside the scope of this study.

(6) The role of wbbY is intriguing, pointing towards the importance of high molecular weight O-polysaccharide. In this mutant background, the authors need to assess whether the expression of the capsule, and ECA is affected. Authors need also to complement the mutant. Which is the mechanism conferring resistance?

We thank the reviewer for their comment and would like to mention that wbbY has already been shown to play a role in LPS profile/biosynthesis and serum-resistance (10.3389/fmicb.2014.00608). Furthermore, blast analysis shows that the wbbY gene between the NTUH-K2044 (strain used in aforementioned study) and ECL8 shares 100% sequence identity and also shares lps operon structure. Hence, we do not find it pertinent to complement this mutant as we believe its molecular mechanism has already been established. We have now in the text more prominently highlighted the results of this study and how our screen was robust enough to also identify this gene for serum resistance.

(7) hns and gnd mutants most likely will have their capsule affected. The authors need to assess whether this is the case. Which is the mechanism conferring resistance?

As mentioned in point 6, we believe that the serum resistance phenotype is attributable to the LPS phenotype. Previous studies have listed hns and gnd mutants would likely have differences in capsule but due to hns being pleiotropic and gnd being intercalated/adjacent to the LPS/O-antigen biosynthesis it would be difficult to exactly delineate which cellular surface structure is involved.

(8) The conclusion section can be shortened significantly as much of the text is a repetition of the results/discussion section.

We thank the reviewer for their suggestion and have made edits to limit repetition in the conclusion section.

**Reviewer #3 (Public Review):**
Below I include several comments regarding potential weaknesses in the methodology used:The study was done with biological duplicates. In vitro studies usually require 3 samples for performing statistical robust analysis. Thus, are two duplicates enough to reach reproducible results? This is important because many genes are analyzed which could lead to false positives. That said, I acknowledge that genes that were confirmed through targeted mutagenesis led to similar phenotypic results. However, what about all those genes with higher p and q values that were not confirmed? Will those differences be real or represent false positives? Could this explain the differences obtained between this and other studies?

We thank the reviewer for their comment and apologize for the confusion, data were only pooled for the statistical analysis of gene essentiality. Here, two technical replicates of the input library were sequenced and the number of insertions per gene quantified (insertion index scores). These replicates had a correlation coefficient of r2 = 0.955, and the insertions per gene data were pooled to give total insertions index scores to predict gene essentiality. For conditional analyses (growth in urine or serum), replicate data were not combined. As mentioned previously, differences between this and other studies could also be attributed to inherent genomic differences or due to differences in experimental methodology, computational approaches, or the stringency of analysis used to categorize these genes.

Two approaches are performed to investigate genes required for K. pneumoniae resistance to serum. In the first approach, the resistance to complement in serum is investigated. And here a total of 356 genes were identified to be relevant. In contrast, when genes required for overall resistance to serum are studied, only 52 genes seem to be involved. In principle, one would expect to see more genes required for overall resistance to serum and within them identify the genes required for resistance to complement. So this result is unexpected. In addition, it seems unlikely that 356 genes are involved in resistance to complement. Thus, is it possible false positives account for some of the results obtained?

We thank the reviewer for their comment and do believe false positives may account for some of the identified genes. Specifically, to the large contrast in genes, we believe this is due to the methodology as alluded to in our conclusion section. For overall resistance to serum, we used a longer time point (180 min exposure) where fewer surviving mutants are recovered hence fewer overall genes will be identified, whereas strains with short killing windows will have more (i.e. complement-mediated killing, 90 minute exposure).

**Reviewer #3 (Recommendations For The Authors):**
In Figure 4 it is shown that genes important for growth in urine include several that are required for enterobactin uptake. Moreover, an in vitro experiment shows that the complementation of urine with iron increases K. pneumoniae growth. It would have been informative to do a competition experiment between the WT and Fep mutants in urine supplemented with iron. This could demonstrate that the genes identified are only necessary for conditions in which iron is in limiting concentrations and confirm that the defect of the mutants is not due to other characteristics of urine.

We appreciate this suggestion. We have now included a new panel (Figure 5G) addressing the supplementation of iron in urine for these select mutants.

Considering the results section, the title for Figure 6 seems to be more appropriate for Figure 5.

Thank you, this has now been corrected.

Other points:Line 44: treat instead of treating

Thank you, this has now been corrected.

Line 63: found that only 3 genes played a role instead of "found only 3 genes played a role"

Thank you, this has now been corrected.

Line 105: is there any reason for only using males? Since UTIs are frequent in women? Why not use urine from women volunteers?

Due to accessibility of willing volunteers and human ethic application processes, only male samples were available. We are currently undertaking further studies to understand how male and female urine influences growth of uropathogens.

Line 105: since the urine was filter-sterilized, maybe the authors can comment that another point that is missing in urine - and that it may be important to study - will be the presence of the urine microbiome and how this affects growth of K. pneumoniae.

We again thank the reviewer for this comment and have now edited the manuscript discussing how the absence of urine microbiome could affect growth (Line 659). As an aside, future studies in our lab are interested in looking at the role of commensal/microbiome co-interactions for essentiality/pathogenesis using TraDIS.

Line 116: I understand that the 8 healthy volunteers combined males and females

Thank you, we have now edited this methods line to make this clearer.

Line 120: incubate in serum 90 min and 180 RPM shaking: any reasons for using these conditions, any reference supporting these conditions?

Thank you for pointing this out, we were mirroring a previous K. pneumoniae serum-resistance study (doi.org/10.1128/iai.00043-).

Line 156: space after the dot.

Thank you, we have now corrected this in the manuscript.

Line 164: resulting reads were mapped to the K. pneumoniae: what are the parameters used for mapping (e.g. % of identity...)?

Thank you for bringing this to our attention, we have now included in our manuscript that we used the default parameters of BWA-MEM for mapping for minimum seed length (default -k = 20bp exact match)

Line 180: it will be good to upload to a repository the In-house scripts used or indicate the link beside the reference for those scripts.

Our scripts are derived from the pioneering TraDIS study (doi: 10.1101/gr.097097.109). We are currently still optimizing our scripts and intend to upload these to be publicly available. However, in the meantime we are more than happy to share them with other parties upon request.

Line 191: why were genes classified as 12 times more likely to be situated in the left mode? Any particular reason for using this threshold?

We opted for a more-stringent threshold for classifying essential genes, in keeping with previous and comparable studies (doi.org/10.1371/journal.pgen.1003834).

Line 209: do you mean Q-value of <0.05 instead of >0.05 ? How is this Q value is calculated, and which specific tests are applied?

Thank you for pointing out this Q value error, we have now corrected this in the manuscript. These values were generated using the biotradis tradis_comparison.R script which uses the EdgeR package. For further reading please see DOI: 10.1093/bioinformatics/btp616. The Q-values are from P values corrected for multiple testing by the Benjamini-Hochberg method.

Line 212: again, which type of test is used? What about the urine growth analysis? The same type of tests were applied?

Thank you for bringing this to our attention, we have now indicated in the referenced method section the use of which package for which datasets (i.e. or serum). Line 212 refers to our use of the AlbaTraDIS package, which builds on the biotradis toolkit, to identify gene commonalities/differences in the selected growth conditions again using multiple testing by the Benjamini-Hochberg methods. For further reading, please refer to DOI: 10.1371/journal.pcbi.1007980

Line 226: do the authors mean Sanger sequencing instead of SangerSanger sequencing?

Thank you, we have now corrected this in the manuscript.

Line 239: does the WT strain contain another marker for differentiating this strain from the mutant? Or is the calculation of the number of WT CFUs done by subtracting the number of CFUs in media with antibiotics from the total number of CFUs in media without antibiotics? The former will be a more accurate method.

The calculation was based on the latter assumption, “number of WT CFUs done by subtracting the number of CFUs in media with antibiotics from the total number of CFUs in media without antibiotics”. We have now updated the methods section to make this clearer.

Line 266: can you indicate approximately how many CFUs you have in this OD?

Thank you, we have now also indicated an approximate CFU for this mentionedOD600 (OD600 1 = 7 × 108 cells).

Line 309: besides indicating Figure 1D please indicate here Dataset S1 (the table where one can see the list of essential and non-essential genes). This table is shown afterwards but I think it will be more appropriate to show it at the begging of the section.

Thank you, we have now taken on this recommendation and have now edited the manuscript to also indicate Dataset S1 earlier.

Table 3. regarding the comparison of essential genes between different strains. I think it will be more clear if a Venn diagram was drawn including only genes that have homologs in all the studied strains (i.e. defining the core genome essentially).

We would like to thank the reviewer for suggesting a venn diagram and have now removed Table 3 which has been replaced with a new Figure 3.

Line 461: replicates were combined for downstream analyses? But are replicates combined for doing the statistical analysis? If so, how is the statistical analysis performed? How is it taken into account the potential variability in the abundance in each library? An r of 0.9 is high but not perfect.

Technical replicates of the sequenced input library were combined following identification of a correlation coefficient of r2 = 0.955, for the calculation of insertion index scores used in gene essentiality analysis. While r2 = 0.955 is not perfect, discrepancies here can be attributed to higher variance in insertion index scores when sampling small genes, as these are represented by fewer insertions and the stochastic absence of a single insertion event has a greater effect on the overall IIS. Replicate data were not pooled for statistical analysis of mutant fitness (growth in urine and serum).

Line 487: is there any control strain containing the kanamycin gene in a part of the genome that does not affect the growth of K. pneumoniae? This could be used to show that having the kanamycin gene does not provide any defect in urine growth.

We thank the reviewer for this suggestion but argue that introduction of the kanamycin gene into each unique loci may result in various levels of gene fitness that would be incomparable to a single control strain. Instead, we culture the ECL8 mutant library in urine and ensure that its kinetics are comparable to the wildtype. As the library contains thousands of kanamycin cassettes uniquely positioned across most of the genome with no observable growth defect, we do not anticipate the presence or expression of the cassette to have an appreciable impact.

Line 569: in the methodology it was indicated that control cells were incubated in PBS for the same amount of time. I think this is an important control that is not cited in the results section. Please can you indicate?

We apologise for this misunderstanding due to how the methodology was written. The experiment did not sequence the PBS incubated samples as this was solely used a check for viability of the used K. pneumoniae ECL8 stock solution.

Line 597: "Mutants in igaA are enriched in our experiments". Can you show this data?

We have now included this as a supplementary (Figure S11A)

Line 615: when doing this calculation, I guess the authors take into account only genes that are also present in the other strains.

That is correct, we were aiming to highlight the high conservation of “essential genes” among all the selected strains.

Line 627: why surprisingly? Because is too low. Then indicate.

Thank you, we have now edited this sentence to indicate that.

Figure 4: please, for clarity, can you indicate the meaning of the colors in the figure itself besides indicating it in the figure legend?

Thank you, we have now included a color legend in these figure panels for clarity.